# A genome-wide relay of signalling-responsive enhancers drives hematopoietic specification

B. Edginton-White[1,4] ✉, A. Maytum[1,4], S. G. Kellaway [ID][1], D. K. Goode[2], P. Keane [ID][1], I. Pagnuco[3,1], S. A. Assi[1], L. Ames[1], M. Clarke[1], P. N. Cockerill [ID][1], B. Göttgens [ID][2], J. B. Cazier[1,3] & C. Bonifer [ID][1] ✉

Developmental control of gene expression critically depends on distal cis-regulatory elements including enhancers which interact with promoters to activate gene expression. To date no global experiments have been conducted that identify their cell type and cell stage-specific activity within one developmental pathway and in a chromatin context. Here, we describe a high-throughput method that identifies thousands of differentially active cis-elements able to stimulate a minimal promoter at five stages of hematopoietic progenitor development from embryonic stem (ES) cells, which can be adapted to any ES cell derived cell type. We show that blood cell-specific gene expression is controlled by the concerted action of thousands of differentiation stage-specific sets of cis-elements which respond to cytokine signals terminating at signalling responsive transcription factors. Our work provides an important resource for studies of hematopoietic specification and highlights the mechanisms of how and where extrinsic signals program a cell type-specific chromatin landscape driving hematopoietic differentiation.

The blueprint for the developmental regulation of gene expression is encoded in our genome in the form of cis-regulatory elements that exist as nuclease hypersensitive sites in chromatin. These elements are scattered over large distances and integrate multiple intrinsic and extrinsic signals regulating the activity of transcription factors (TFs) that bind to such elements[1,2]. TFs and TF encoding genes together with their targets form gene regulatory networks (GRNs) that define the identity of a cell. TFs together with chromatin remodellers/modifiers form multi-molecular complexes that assemble on cis-regulatory elements and interact with each other within intranuclear space to activate gene expression[2]. To answer the question of how one GRN transits into another in development, it is essential (i) to identify and characterize the full complement of cell type and cell stage-

specific transcription regulatory elements, (ii) to identify the TFs binding to them and (iii) to understand how they respond to external cues.

For several decades, reporter gene assays have been used to define cis-regulatory elements as enhancers or promoters, with enhancers being able to increase transcription from promoters independent of their orientation[3,4]. However, assessing enhancer activity within a chromatin context is more difficult. Studies inserting individual enhancer-promoter combinations at different genomic locations revealed that the local chromatin environment strongly influences gene expression. Such effects can typically only be overcome if the full complement of cis-regulatory elements is present on a transgene, making the analysis of individual elements difficult as their deletion make transgenes again

[1]Institute of Cancer and Genomic Sciences, School of Medicine and Dentistry, University of Birmingham, B152TT Birmingham, UK. [2]Department of Haematology, Wellcome and Medical Research Council Cambridge Stem Cell Institute, Jeffrey Cheah Biomedical Centre, Cambridge Biomedical Campus, University of Cambridge, Cambridge CB2 0AW, UK. [3]Centre for Computational Biology, Institute of Cancer and Genomic Sciences, University of Birmingham, B152TT Birmingham, UK. [4]These authors contributed equally: B. Edginton-White, A. Maytum. ✉e-mail: B.Edginton-White@bham.ac.uk; c.bonifer@bham.ac.uk

susceptible to position effects as reviewed in[5]. The deletion of individual elements within a gene locus can uncover enhancer function, but often misses developmental stage-specific elements, at least in part due to functional redundancy with neighbouring elements. Therefore, multiple surrogate markers have been identified that correlate with a high activity of the gene linked to the respective cis-regulatory element, including DNaseI hypersensitivity, TF binding, histone acetylation/mono-methylation and enhancer transcription[3,6–11]. None of these features, alone or in combination was fully predictive of enhancer activity with TF binding being the best predictor[12,13]. Consequently, functional assays remain essential to ascertain whether any given element can stimulate transcription in a chromatin context.

During embryonic development, definitive blood cells including hematopoietic stem cells develop from mesoderm derived endothelial cells within the dorsal aorta[14,15]. The specification of hematopoietic cells in the embryo and hematopoietic cell differentiation in the adult have served as an important model to reveal general principles of the control of gene expression in mammalian development[16]. The roles of the most important TFs together with signals such as cytokines controlling different developmental stages are known, and most intermediate cell types have been identified. Moreover, in vitro differentiation of human and mouse embryonic stem cells (ESCs) recapitulates embryonic hematopoietic development, thus facilitating deep molecular analysis into the developmentally-controlled transition of gene regulatory networks (GRNs)[17].

We previously reported a multi-omics analysis revealing dynamic GRNs that are specific for each of the major stages of blood cell specification from ESCs. We identified the locations and dynamic activities of sets of cis-regulatory elements associated with developmental gene regulation and based on this data uncovered important pathways, such as Hippo signalling that are required for blood cell formation[18–20]. However, major questions are still open. Whilst we could correlate chromatin alterations with dynamic gene expression[21], our data did not provide functional evidence for which cis-regulatory elements have enhancer activity, how they are controlled at different developmental stages, how extrinsic signals control their activity and importantly, which TFs mediate signalling responsiveness.

In the work presented here, we describe the development of a high-throughput method identifying thousands of enhancer and promoter elements specifically active at defined stages of blood cell specification in a chromatin environment and correlate their activity with gene expression in the same cells. We are using mouse ESCs to be able to integrate our results with our previously global multi-omics collected data, but the method can be expanded to any cell type that can be differentiated from ESCs and can easily be adapted to human ESCs. We show that the same elements exist as active chromatin in vivo in the appropriate mouse cell types. Finally, we identify cytokine responsive enhancer elements, and for one cytokine, VEGF, characterize the TFs mediating its activity within the GRN driving blood cell development. Our work provides a tool to significantly advance our understanding of developmental gene expression control in the hematopoietic system and beyond.

## Results

### Establishing a high-throughput method for functional enhancer testing in a chromatin environment

We identified functional enhancer elements in the chromatin of mouse embryonic stem cells and their differentiated progeny representing different stages of hematopoietic specification (Fig. 1a)[18,19]. The first stage analysed here consists of FLK1-expressing hemangioblast-like cells[22] which have the ability to differentiate into cardiac, endothelial and hematopoietic cells and which are purified from embryoid bodies (EBs) after day 3 of culture. These cells are then placed into blast culture and form a

mixture of (i) hemogenic endothelium 1 (HE1) which expresses a low level of RUNX1, (ii) hemogenic endothelium 2 (HE2) cells which up-regulate RUNX1 and CD41 but are still adherent and (iii) blast-like hematopoietic progenitor cells (HP) that underwent the endothelial-hematopoietic transition (EHT) and float off into the culture medium. Using the differential expression of specific surface markers (KIT, TIE-2 and CD41) each cell type can be purified to near homogeneity.

Figure 1 and Supplementary Fig. 1 show the enhancer identification pipeline which is based on the system developed by Wilkinson et al. 2013[23,24] and was adapted for a high-throughput genome-wide screen. Essentially, we differentiate cells, sort them into different developmental stages as shown in Fig. 1a, purify ATAC-Seq fragments for each differentiation stage, clone them into a targeting vector to generate a fragment library which is then integrated into a defined target site in the *HPRT* locus carrying a minimal promoter to drive a reporter gene (Venus-YFP) (Fig. 1b). By employing this targeting system, we ensure that only one fragment and reporter construct is active in each cell and that it is in an accessible chromatin environment throughout differentiation. We then differentiate cells and purify cells from each stage of development for reporter activity measurements (Supplementary Fig. 1a, b). Cell populations expressing high, medium and low YFP levels are isolated together with YFP negative cells (Supplementary Fig. 1b). Fragment inserts are sequenced after amplification using barcoded primers recognising the ATAC linkers. Each unique aligned read in the sequencing data is representative of a fragment that was cloned into the reporter construct and is assigned activity based on which FACS (YFP) population it was sorted into. Sequences then undergo a rigorous filtering against different criteria as detailed in Supplementary Fig. 1c (discussed in more detail in Supplementary Notes). The original plasmid libraries cover between 90% and 98% of all ATAC-Seq peaks (Supplementary Fig. 1d). Our screen was conducted in two replicates identifying several hundred-thousand fragments with transcription-stimulatory activity (Supplementary Fig. 1f). 22–31% of fragments were located within annotated distal elements (Supplementary Fig. 1d), covering more than 70,000 enhancer-positive ATAC sites across all stages (Fig. 1c left panel, Supplementary Fig. 1e, Supplementary Dataset 1). The remaining fragments were promoter sequences which were classified as being within 1.5 kb of an annotated transcription start site (Fig. 1c, right panel, Supplementary Fig. 1e). An example for enhancer annotation is shown in Fig. 1d, depicting the well-characterized Spi1 (PU.1) locus, which captures all previously identified enhancer elements together with their known stage-specific activity[25] with other examples shown in Supplementary Notes. Most ATAC-fragments displaying stimulatory activity in our reporter assay (see scheme in Supplementary Fig. 2a, Supplementary Fig. 1e) overlapped with fragments within the same ATAC site that did not score, indicating that the vast majority of captured open chromatin regions can regulate transcription. Moreover, between 30% and 50% of all distal ATAC sites and around 80% of all promoter sites in the ATAC library contained a fragment scoring in our assay (Fig. 2a; Supplementary Fig. 2b). The median number of positive fragments per distal ATAC site was between 3 and 6 (Supplementary Fig. 1g).

We next integrated our enhancer data with previously published chromatin immunoprecipitation (ChIP)-Seq data characterizing histone modifications at cis-regulatory elements in the same experimental system[18]. About 30% to 60% of all enhancer sites overlap with H3K27Ac regions (Fig. 2b, Supplementary Fig. 2c). We find a significant overlap of our positive but not our negative/unknown enhancer ATAC sites with the VISTA enhancer database which describes 1061 functionally identified enhancers[26] (Fig. 2c, Supplementary Fig. 2d). The size distribution of positive and negative (non-scoring) fragments was the same (Supplementary Fig. 2e) indicating an absence of size selection for active fragments. Most enhancer positive ATAC fragments

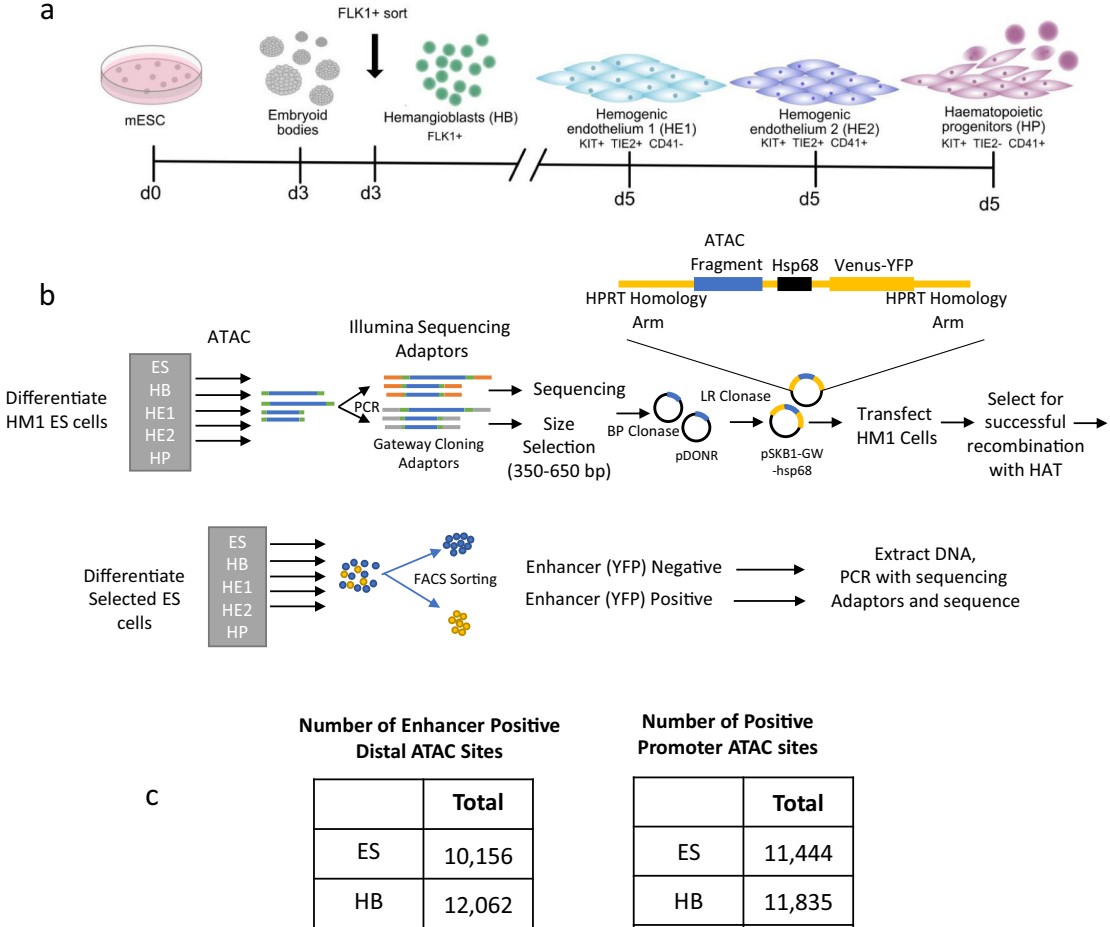

Number of Enhancer Positive Distal ATAC Sites

|  | Total |
|---|---|
| ES | 10,156 |
| HB | 12,062 |
| HE1 | 18,180 |
| HE2 | 16,891 |
| HP | 13,323 |

Number of Positive Promoter ATAC sites

|  | Total |
|---|---|
| ES | 11,444 |
| HB | 11,835 |
| HE1 | 12,240 |
| HE2 | 12,087 |
| HP | 11,633 |

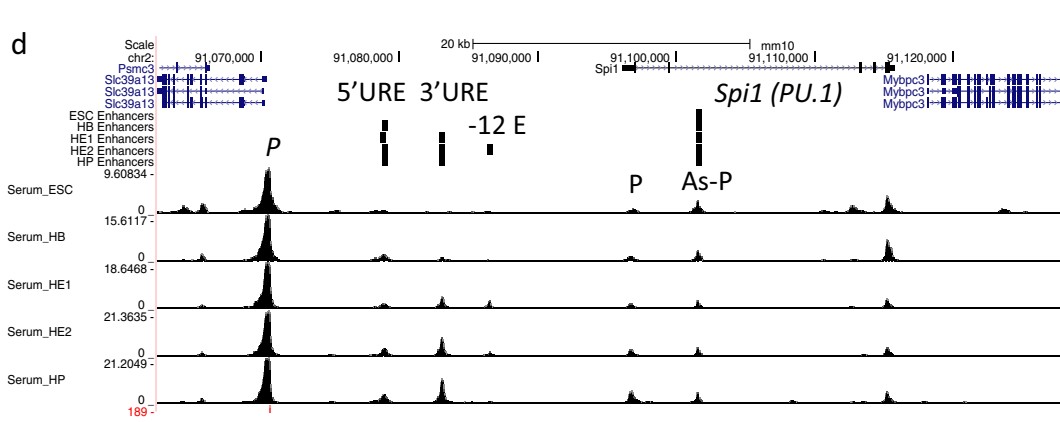

**Fig. 1 | Establishing a high-throughput method for functional enhancer testing in a chromatin environment. a** Depiction of the ES cell differentiation system and the cell types analysed. **b** Overview of the screening procedure using ATAC-Seq fragments from 5 different cell stages. **c** Total number of distal and promoter ATAC-sites scoring positive in two replicates of the assay. **d** UCSC Browser screenshot showing an example of well characterized enhancers (5′URE, 3′URE, −12 kb enhancer, promoter (P) and antisense promoter (As-P) from the SPI1 (PU.1) locus[25] marked by ATAC-Seq peaks which were recovered in our screen. Distal cis regulatory elements active at the different stages scoring positive in our assay (enhancers) are indicated by vertical bars.

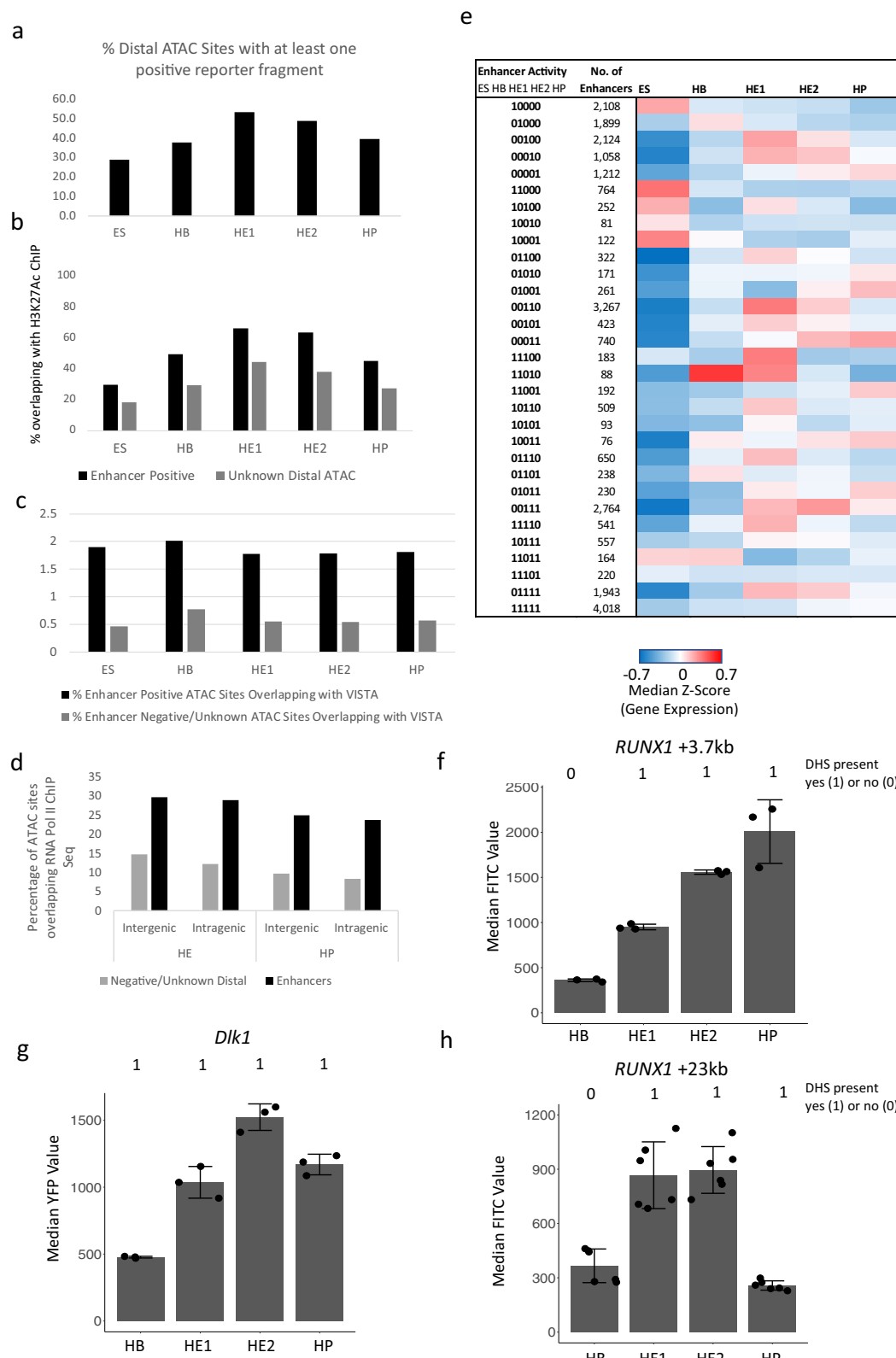

overlap with open chromatin sites found in purified hemogenic endothelium and endothelial cells from day 9.5 and day 13.5 mouse embryos[27,28] indicating that they are active in vivo (Supplementary Fig. 2f). Finally, the comparison with previously collected TF ChIP data[18,20,29–31] shows that between 17% (in HB) and 76% (in HP cells) of enhancer fragments are bound by ubiquitous and differentiation

stage-specific TFs, depending on the number of available ChIP experiments for each stage.

It was reported that active enhancers are bound by RNA-Polymerase II (Pol II) and are transcribed[8,32]. To examine the correlation between the ability of distal elements identified in our study to drive RNA production during development we examined whether they

**Fig. 2 | Characterization of enhancer features and association of cell stage-specific enhancer activity with cell stage-specific gene expression. a** Distal ATAC sites containing at least one fragment scoring positive in our assay. Source data are provided as a Source Data file. **b** Percentage of functionally identified distal elements scoring positive in our screen overlapping with histone H3 lysine 27 (H3K27Ac) peaks as compared to inactive elements or where activity is not known. ChIP data from ref. [18]. Source data are provided as a Source Data file. **c** Overlap of fragments scoring positive in our assay with known enhancers from the VISTA database[26]. Source data are provided as a Source Data file. **d** Percentage intergenic and intragenic fragments scoring positive in our assay overlapping with RNA polymerase II binding sites in the hemogenic endothelium and HP cells. Pol II data from ref. [30]. Source data are provided as a Source Data file. **e** Presence or absence of enhancer activity at the different developmental stages expressed as binary code (0 = scoring negative; 1 = scoring positive) and sorted by the activity pattern across

differentiation. The heatmap depicts the activity of the genes associated with these elements. Enhancer - promoter association was determined by using the union of HiC data from ES and HPC7 (HP) cells[29] together with co-regulation data determined by ref. [21] in a total of 21,671 elements. All promoters not covered by these data (5599) were associated with being the nearest to the enhancer element. **f–h** Activity profiles of individual enhancers identified in our screen in single ES cell clones during differentiation. The presence or absence of an ATAC-peak is indicated by the binary code used in (**e**). **f** *RUNX1* + 3.7 enhancer element (chr16:92822182-92822601). **g** *Dlk1* enhancer element (Chr12: 109437601-109438085), **h** *RUNX1* + 23 kb enhancer element[49]. Data are presented as mean values + /− standard deviation (SD). Dots showing individual values for *Dlk1* and *Runx1* + 3.7 kb *n* = 3 biologically independent experiments and for *Runx1* + 23 kb *n* = 6 biologically independent experiments. For sequence details see Supplementary Notes. Source data are provided as a Source Data file.

were capable of binding Pol II using a previously published data-set from ESC generated HE and HP cells[30]. Figure 2d shows that up to 30% of identified enhancers are associated with Pol II binding.

## Sequential stages of hematopoietic specification are defined by distinct enhancer sets

The genomic information for tissue-specific gene expression manifests itself in the activity pattern of distal regulatory elements[18,33]. An important feature of our method is therefore the identification of developmental stage-specific enhancer activity (Supplementary data-set 2). Between 10% and 20% of all distal and between 15% and 50% of all promoter sites showed stage specific activity in our assay (Supplementary Fig. 3a). To assign genes to regulatory elements, we used publicly available HiC and co-regulation data[21,29,34] and examined how gene expression correlated with enhancer activity by generating a binarized matrix cataloguing enhancers as active (1) and inactive (0) at each of the five differentiation stages (Fig. 2e, Supplementary data-sets 2, 3). This analysis which depicts gene expression at the different stages in a heatmap shows that (i) enhancer activity during development is largely continuous and (ii) stage-specific gene expression is strongly associated with stage specific enhancer activity. Examples for stage-specific active enhancers can be found in Fig. 2f–h and Supplementary Fig. 2g.

We then determined for each differentiation stage, which TF motif combinations were associated with enhancer activity (Fig. 3 and Supplementary Fig. 3). We first examined the motif content of distal ATAC-Seq sites at specific differentiation stages by defining distal elements specific for each stage[18]. In line with previously published ChIP data[18,20,29], cell stage-specific cis-element patterns are associated with stage-specific TF motifs (Fig. 3a), with those for hematopoietic TFs such as RUNX1 or PU.1 being enriched in HP cells and those for the HIPPO signalling mediator TEAD and SOX factors enriched in the HE. This pattern was also seen with stage-specifically active enhancer but not with promoter fragments (Supplementary Fig. 3b). In contrast, ubiquitously active enhancer fragments displayed a similar motif signature at all developmental stages, reinforcing the notion that tissue-specificity is encoded in distal elements (Supplementary Fig. 3c, left panel). The motif for the Zn++ finger factor CTCF was enriched in active distal enhancer but not promoter fragments. It was recently shown in differentiating erythroid cells that dynamically bound CTCF cooperates with lineage-specific TFs bound to distal elements to interact with promoters[35]. Our data are consistent with this finding.

We next asked whether stage-specific enhancer activity was correlated with a pattern of cooperating TFs. To this end, for each developmental stage we performed a motif co-localization analysis which examines whether specific binding motif pairs located within 50 bp of each other were enriched in stage-specific enhancer fragments as compared to all open chromatin sites (analysis scheme depicted in Supplementary Fig. 3d). Motifs with a 100% overlap were

removed. Stage-specific enhancer activity correlated with enriched colocalizations of motifs for developmental-stage specific TFs (Fig. 3b–f). In line with their generally open chromatin structure[36], ESC-specific enhancers showed a great variety of paired motifs such as those for the pluripotency factors NANOG/SOX2/OCT4. Interestingly, AP-1 motifs showed a very high co-occurrence both with itself and other motifs, potentially linking enhancer activity to signalling processes (Fig. 3b). AP-1 homo-typic motif associations were also found at HB-specific enhancer fragments but co-localization with pluripotency factor motifs was lost (Fig. 3c). The HE1 stage showed a strong enrichment in RBPJ, AP-1 and SMAD motif pairs, with AP-1 motifs colocalizing with high frequency with most other factors (Fig. 3d). The number of HE2-specific enhancers was low as it is a transitory stage. Here, PU.1 motifs show increased co-localization with SMAD and OCT motifs (Fig. 3e). This observation is well-supported by studies that TGFβ and BMP-induced SMADs often follow lineage-specific master transcription factors to cell stage-specific enhancers[37,38]. As expected from previous ChIP studies[39], we find a co-localization of motifs for hematopoietic TFs such as C/EBP, RUNX, PU.1 and GATA at the HP-stage. AP-1 motifs again co-localized with a variety of other motifs, including C/EBP, RUNX1 and PU.1 (Fig. 3f). Although generally enriched in enhancer fragments at all stages (Supplementary Fig. 3c), CTCF motifs were not significantly paired with any other motif.

## Identification of cytokine-responsive enhancer elements

Cell differentiation involves extracellular signals which alter growth and differentiation states by changing gene expression. Signalling molecules such as cytokine receptors, integrins and kinase molecules are well characterized, but less is known of how different signals are integrated at the level of the genome. In spite of a number of efforts looking at specific genomic regions such as[40,41], we have limited global information about which cis-regulatory elements can respond to signals, which TF combinations are involved and how they cooperate to ensure that the genome responds to outside signals in a coordinated and balanced fashion. Our global cis-element collection allows us to answer these questions.

To this end, we employed a serum-free in vitro differentiation system[42] that is based on the sequential addition of growth factors such as BMP4, VEGF and hematopoietic cytokines. We tested how individual cytokines affected the differentiation profile and the open chromatin landscape of HE1, HE2 and HP cells sorted as described in Fig. 4a. The comparison of the chromatin signature of cells differentiated in serum and under serum-free conditions (Supplementary Fig. 4a), showed that around 80−90% of all promoters active in cells from serum-free culture overlapped in both conditions, demonstrating the reproducibility of our differentiation system. However, whilst the overlap for HB and HE1 cells was high (>70%) for HP cells we noticed changes in the bulk open chromatin landscape which only affected the distal elements. This result indicates that although the cellular identity seemed to be largely preserved in sorted cells

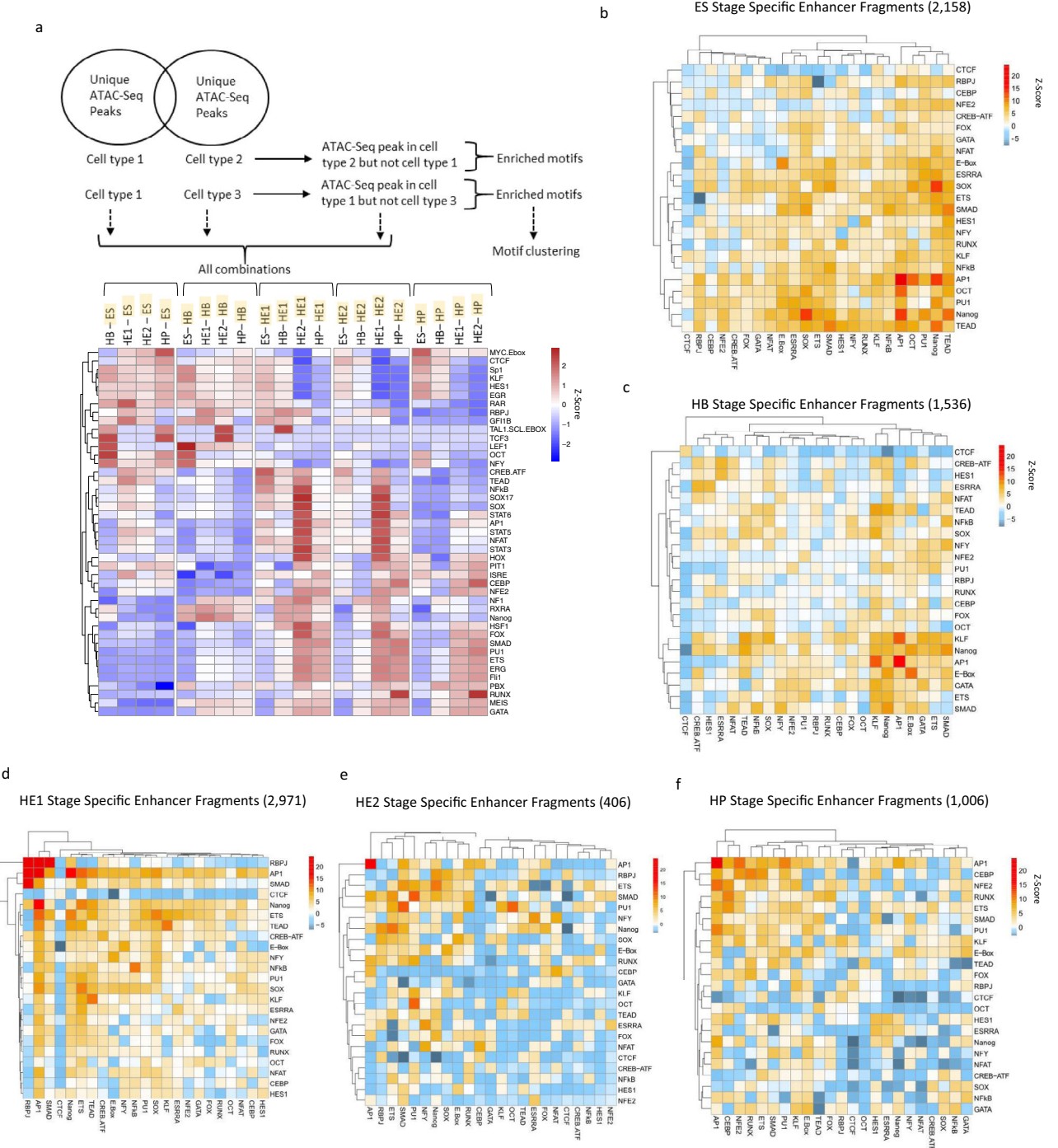

**Fig. 3 | Stage-specific enhancer elements show a specific TF motif and motif co-localization pattern. a** Identification of specific TF binding motifs in cell type specific open chromatin regions (distal elements only) as described in ref. [18]. The upper panel shows the filtering strategy. The lower panel shows how specific binding motifs are enriched in ATAC-Sites that are specific for each cell type (highlighted in yellow) as compared to another cell type (indicated by brackets). **b**–**f** Motif co-localization analysis. Heatmaps depicting Z-Score enrichments for pairs of TF binding motifs within 50 bp for five cell differentiation stages as indicated on top of each heat-map, together with the number of fragments analyzed.

Binding motifs for the indicated TFs are listed on the right and the bottom of the heat-map. Position-weight matrices used for this analysis can be found in Supplementary Notes Table 1. The overall strategy is outlined in Supplementary Fig. 3d. Note that we do not expect the diagonal to be a uniform line. In the heatmap the diagonal represents the tendency for a motif to co-localize with itself which only yields a high score with groups of multiple of the same motif within 50 bp of each other, and if these groups occur more frequently compared to the background peak set. This feature will be different for each motif.

expressing the right combination of surface markers, the difference in the signalling environment exerted a strong effect on the chromatin landscape.

To elucidate the role of different cytokines in chromatin programming, we differentiated cells in the presence and absence of

BMP4, VEGF, IL-6 and IL-3, respectively, sorted the different cell types and examined which open chromatin regions changed at least two-fold in response to cytokine removal (Fig. 4a). In total, more than 10,000 unique open chromatin regions were up or down regulated (Supplementary Fig. 4b) which included both distal elements and promoters.

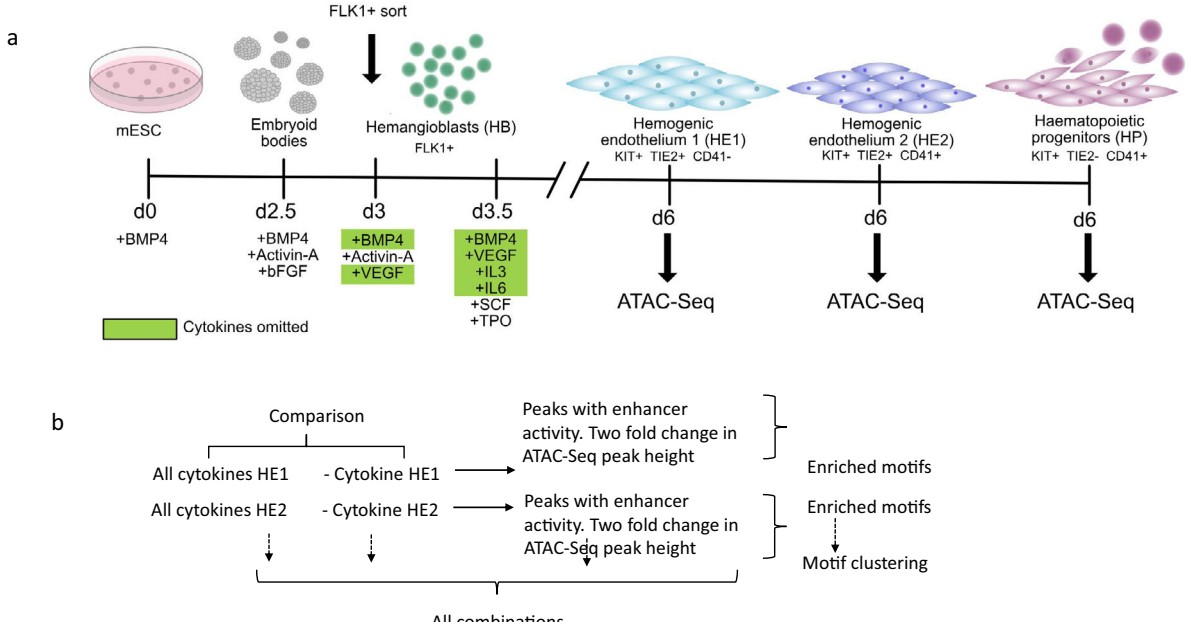

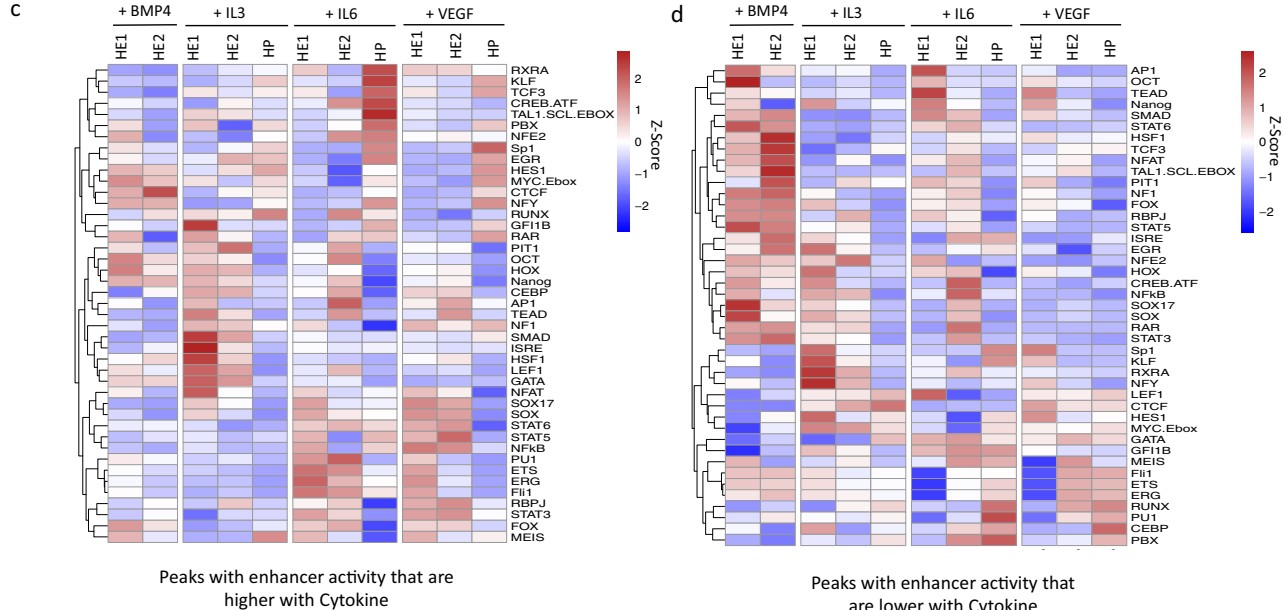

**Fig. 4 | Identification of cytokine -responsive enhancer elements. a** Overview of serum-free in vitro blood progenitor differentiation modified from[42]. Specific cytokines that were left out at the beginning of blast culture are highlighted in green. **b–d** Identification of cytokine responsive distal elements scoring positive in our assay in the presence or absence of the indicated cytokines. **b** Overview of motif

enrichment analysis strategy, **c**, **d** TF binding motif enrichment in distal ATAC peaks containing fragments scoring positive in our assay that are increased (**c**) or decreased (**d**) at least 2-fold in the presence of the indicated cytokines. Position-weight matrices used for this analysis can be found in Supplementary Table 1 in Supplementary Notes.

We next examined, which TF binding motifs were enriched in cytokine-responsive ATAC-Seq peaks with enhancer activity, i.e., peaks that were at least 2-fold higher or lower in the presence of cytokines (Fig. 4b–d). Relative enrichment scores were calculated for each binding motif which were transformed to a Z-Score, hierarchically clustered and plotted as a heatmap (see methods). The alteration of cytokine conditions had a profound influence on chromatin pro-gramming (Fig. 4c, d). The absence of BMP4 from the blast culture

onwards was incompatible with HP formation and open chromatin regions with enhancer activity containing SMAD, HOX, RAR and NOTCH motif signatures were lost in HE (Fig. 4d). This finding is in keeping with these factors being required to form the HE[43,44]. The presence of VEGF led to a loss of peaks with a hematopoietic motif signature in HE2/HP, such as RUNX1, FLI1, GATA and PU.1 motifs. A similar motif enrichment pattern was found when all open chromatin regions were analysed (Supplementary Fig. 4c–e).

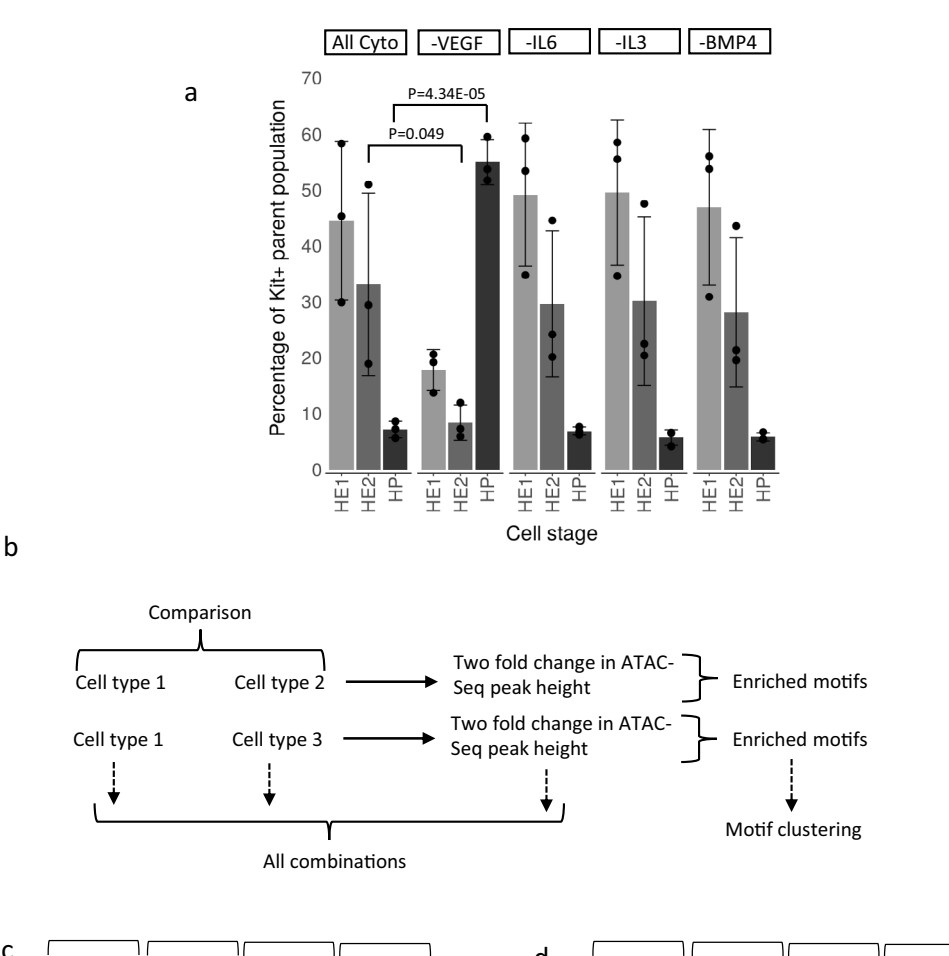

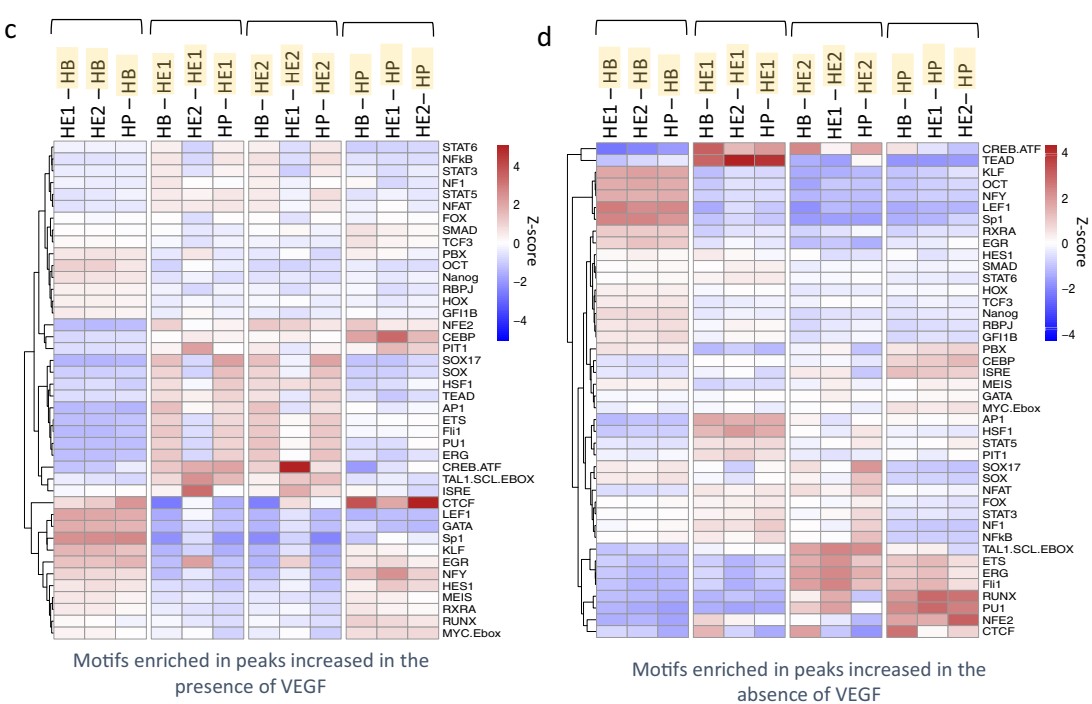

Motifs enriched in peaks increased in the presence of VEGF

Motifs enriched in peaks increased in the absence of VEGF

To assess the effect of cytokine withdrawal on differentiation, we examined the effect of cytokine omission on the frequency of HE1, HE2 and HP cells (Fig. 5a). The presence or absence of BMP4, IL-6 and IL-3 did not influence the proportion of generated HE1, HE2 and HP cells as compared to the all-cytokine condition (Fig. 5a). In contrast, VEGF strongly suppressed hematopoietic progenitor formation.

**The balance between hematopoietic and endothelial development is regulated by VEGF-responsive cis-regulatory elements**
VEGF had the greatest influence on the formation of hematopoietic progenitor cells (Fig. 5a). We therefore examined its role in regulating enhancer activity in more detail. VEGF omission resulted in a significant decrease in the proportion of HE2 cells gained ($P = 0.049$) and a significant increase in the proportion of HP cells formed

**Fig. 5 | The presence of VEGF suppresses multipotent progenitor (HP) development. a** Proportion of HE1/HE2/HP cells within the differentiation culture in the absence of the indicated cytokines as measured by FACS. Data are presented as mean values +/− SD. Dots showing individual values for $n = 3$ biologically independent experiments. *P*-Values were calculated using two-sided Student's *t*-test. Source data are provided as a Source Data file. **b–d** Un-supervised motif clustering analysis examining enrichment of the TF binding motifs as described in Fig.3a, analyzing in all ATAC peaks in two culture conditions. **b** Motif enrichment strategy.

**c** Motifs enriched in peaks of specific cell types (highlighted in yellow) which were at least 2-fold increased in peak height in the All Cytokine Condition, **d** Motifs enriched in peaks from specific cell types (highlighted in yellow) which were at least 2-fold increased in peak height in the VEGF omission condition. TF binding motifs are depicted at the right of the panel. Relative motif enrichment scores Z-score were calculated by columns. Position-weight matrices used for this analysis can be found in Supplementary Table 1 in Supplementary Notes.

($P = 4.34E-05$) at day 3 blast culture. VEGF omission cultures contained a smaller number of cells but an increased proportion of HP cells (Supplementary Fig. 5a) with VEGF withdrawal at around 12–14 h being optimal for the generation of these two cell types (Supplementary Fig. 5b). Moreover, VEGF removal at different time points of blast culture did not influence the proportion of HE1 cells but led to an inverse correlation of HE2 and HP cell numbers (Supplementary Fig. 5b), indicating that this cytokine impacted on the EHT. These results are consistent with previous observations showing that the receptor for VEGF, FLK1, is essential for the formation of blood islands from hemogenic endothelium cells[45] but once hematopoietic cells are formed, cells become dependent on hematopoietic cytokines. However, the molecular basis of this finding, i.e. which genomic events are responsible for this phenomenon has so far been unclear.

Our ATAC-Seq analysis found 7814 chromatin regions carrying enhancer elements that responded to VEGF (Supplementary Dataset 4). To identify VEGF responsive TFs, we repeated the supervised motif clustering analysis that highlighted cell type specific motif enrichments in the presence and absence of VEGF (Fig. 5b–d). HP cells in VEGF cultures maintain an enrichment of motifs for HES1 which is a mediator of NOTCH signalling[46] (Fig. 5c). In contrast, the omission of VEGF activates enhancers with a hematopoietic motif signature with RUNX1 and PU.1 motifs (Fig. 5d). Moreover, ATAC peaks in -VEGF HE1 cells were enriched in TEAD motifs together with binding motifs for factors linked to inflammatory signalling (AP-1, NFkB and CREB/ATF) which have been shown to be important for stem cell development[47]. A similar motif signature in -VEGF cultures was also seen when ATAC sites were directly compared in a pair-wise fashion (Supplementary Fig. 5c–e) with ATAC peaks derived from HE1 cells from +VEGF cultures enriched for TEAD (29.61% of targets, $P = 1e$-193), AP-1 (16.23% targets, $P = 1e$-317) and SOX (41.39% targets, $P = 1e$-696).

## VEGF blocks the upregulation of *RUNX1* at the chromatin and gene expression level

The data shown above demonstrate that VEGF interferes with the EHT and blood progenitor formation. Both processes are crucially dependent on the TF RUNX1[19] which activates hematopoietic genes and represses endothelial genes in cooperation with the TF GFI1[48]. To examine why the EHT is deficient, we examined the chromatin structure of *Runx1* in the presence and absence of VEGF (Fig. 6a). In the presence of VEGF multiple distal DHSs of *Runx1* fail to form. 5 of these elements score in our enhancer assay, including the previously characterized + 23 kb enhancer (2)[49] together with an enhancer at + 3.7 kb (3) (Fig. 2f, h). Of note, several of these elements, such as the + 3.7 kb enhancer are bound by TEAD and AP-1 as well (Supplementary Notes, Fig. 1a).

The results described so far suggested that the reduced ability of VEGF cultures to undergo the EHT was a result of a failure of *Runx1* enhancer activation and a failure of its transcriptional upregulation in HE2. Although we profiled chromatin in FACS purified cells, these alterations could still be caused by shifts in cell composition. We therefore studied VEGF-mediated changes in gene expression in HE1/2 cells at the single cell level (Fig. 6b–e, Supplementary Fig. 6a–e). Without VEGF the overall numbers of HE1/HE2/HP cells were reduced

(Fig. 6b) but the population showed an increased proportion of HP cells, together with an increase in the proportion of smooth muscle cells (Fig. 6b). However, the cellular identity and the overall differentiation trajectory were not altered (Supplementary Fig. 6d). This shift was in concordance with an incomplete down-regulation of endothelial genes such as *Sox17* in HE1 (Avg Log2FC 1.07, P Val Adj < 1.38E-303) and in HE2 (Avg Log2FC 0.49, P Val Adj 2.18E-174) and Tie2 in HE1 (Avg log2FC 0.17, Adj P Val 1.05E-108) and in HE2 (Avg Log2FC 0.09, P Adj Val 3.55E-12) as seen by sc-RNA-seq differential gene expression analysis (Fig. 6c–e). sc-RNA-seq also revealed a lack of upregulation of *Runx1* in HE2 and HP cells cultured with VEGF (Fig. 6e, right panel). The balance between endothelial and hematopoietic gene expression is controlled by SOX17 which represses *Runx1*[50,51]. After the EHT, RUNX1 together with GFI1 binds to *Sox17* and *Notch1* enhancers (Supplementary notes, Figs. 2, 3)[18,48,52,53] forming a feed forward loop driving EHT progression. *Sox17* and *Runx1* are thus expressed in a mutually exclusive fashion with the former being high before the EHT[51] and then being downregulated and the latter being upregulated from a low level during the EHT[52]. This balance shifted after VEGF was omitted (Fig. 6d, e). We therefore conclude that VEGF signalling interferes with the activation of *Runx1* enhancers in the HE, driving gene expression required for the EHT.

To investigate whether VEGF-responsive enhancers were connected to VEGF-regulated genes, we paired each element that required VEGF activation in HE1, HE2 and HP with its respective gene as shown in Fig. 2e. We then integrated this data with their expression as measured in our single-cell experiments (Supplementary Fig. 6e, Supplementary dataset 4). This analysis uncovered that at high-stringency analysis, between 10% and 19% of VEGF responsive enhancers are linked to VEGF responsive up- or down-regulated genes, thus directly linking alteration in chromatin with changes in gene expression. The analysis of gene ontology (GO) terms of VEGF dependent genes highlighted angiogenesis as the top pathway with the VEGF receptor FLK1 (*Kdr*) being a prominent example ($P = 2.88E$-13, Supplementary dataset 4). VEGF-responsive down-regulated genes linked to down-regulated enhancer elements include hematopoietic regulator genes such as *Runx1, Klf2, Jun* and *Elf1*, validating our general approach (Supplementary Dataset 4).

## The VEGF response involves the TEAD – AP-1 axis

We next examined which specific TFs were connected to endothelium-specific gene expression and VEGF responsiveness. Distal ATAC-Seq peaks specific for HE1 cells derived from cultures containing VEGF were enriched for SOX, E-BOX, AP-1 and TEAD TF motifs, indicating a distinct endothelial TF motif signature (Supplementary Fig. 5c). Similarly, distal ATAC-Seq peaks specific for HE2 cells derived from cultures containing VEGF were also enriched for the endothelial TF motifs SOX (36.91% targets $P = 1e$-858), TEAD (48.27% targets, $P = 1e$-366) and AP-1 (21.45% targets, $P = 1e$-543). We therefore examined several VEGF-responsive and endothelial specific enhancer sequences in more detail, which included the *Sparc, Pxn* and *Hspg2* enhancers (Fig. 7a–c, Supplementary Notes, Figs. 4–6). Inspection of the binding motifs of these elements uncovered that they contained SOX, AP-1, TEAD and the NOTCH-signalling responsive factor RBPJ, which is in concordance with the global HE signature (Fig. 5c), but also RUNX1 motifs.

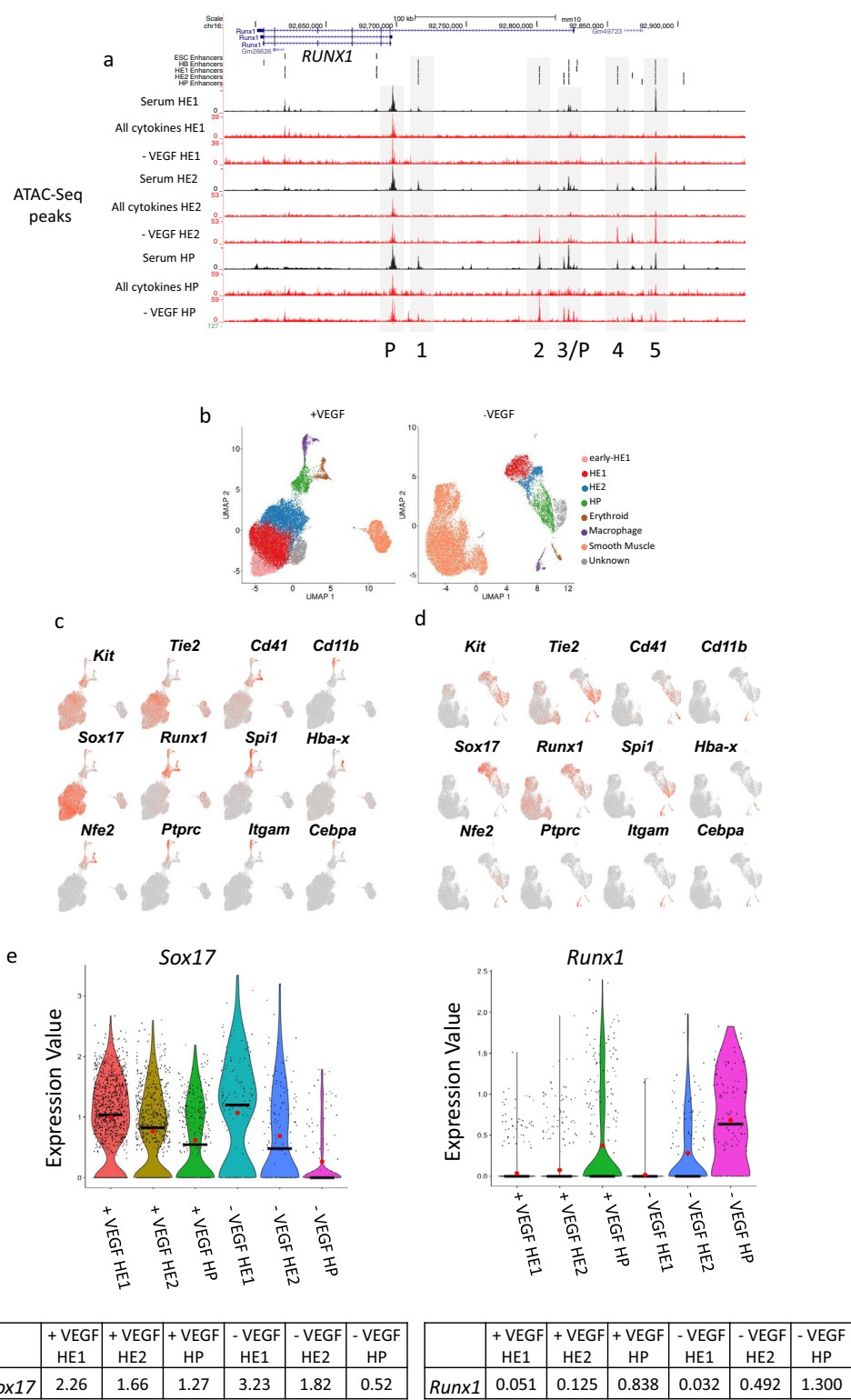

**Fig. 6 | VEGF withdrawal shifts the proportion of endothelial/smooth muscle and hemogenic endothelium/blood progenitor cells by blocking *Runx1* up-regulation. a** UCSC genome browser screenshot depicting open chromatin regions and the locations of distal elements scoring positive in our assay (vertical bars) in the *Runx1* locus under the indicated culture conditions (Serum culture, Serum-free with all cytokines and without VEGF (-VEGF)) and at the different developmental stages. Enhancers (numbered 1–5) and promoter elements (P) are indicated by blue boxes. Enhancer 2 is located at + 23 kb, enhancer 3 is at + 3.7 kb. **b** Single cell gene expression analysis with purified HE1 and HE2 populations in the

presence (+ VEGF) and absence of VEGF (-VEGF). UMAP clustering analysis was performed after all data were pooled. **c, d** Gene expression levels of the indicated genes with (**c**) and without VEGF (**d**) are projected on the clusters. **e** Violin plots of expression values from 6 clusters for *n* = 2266 cells (black line representing the median, red dot the mean and black dots the spread of the data) and mean normalized (Seurat log normalized) gene expression levels of *Sox17* and *Runx1* in the indicated cell types in the presence and absence of VEGF calculated from single cell expression data.

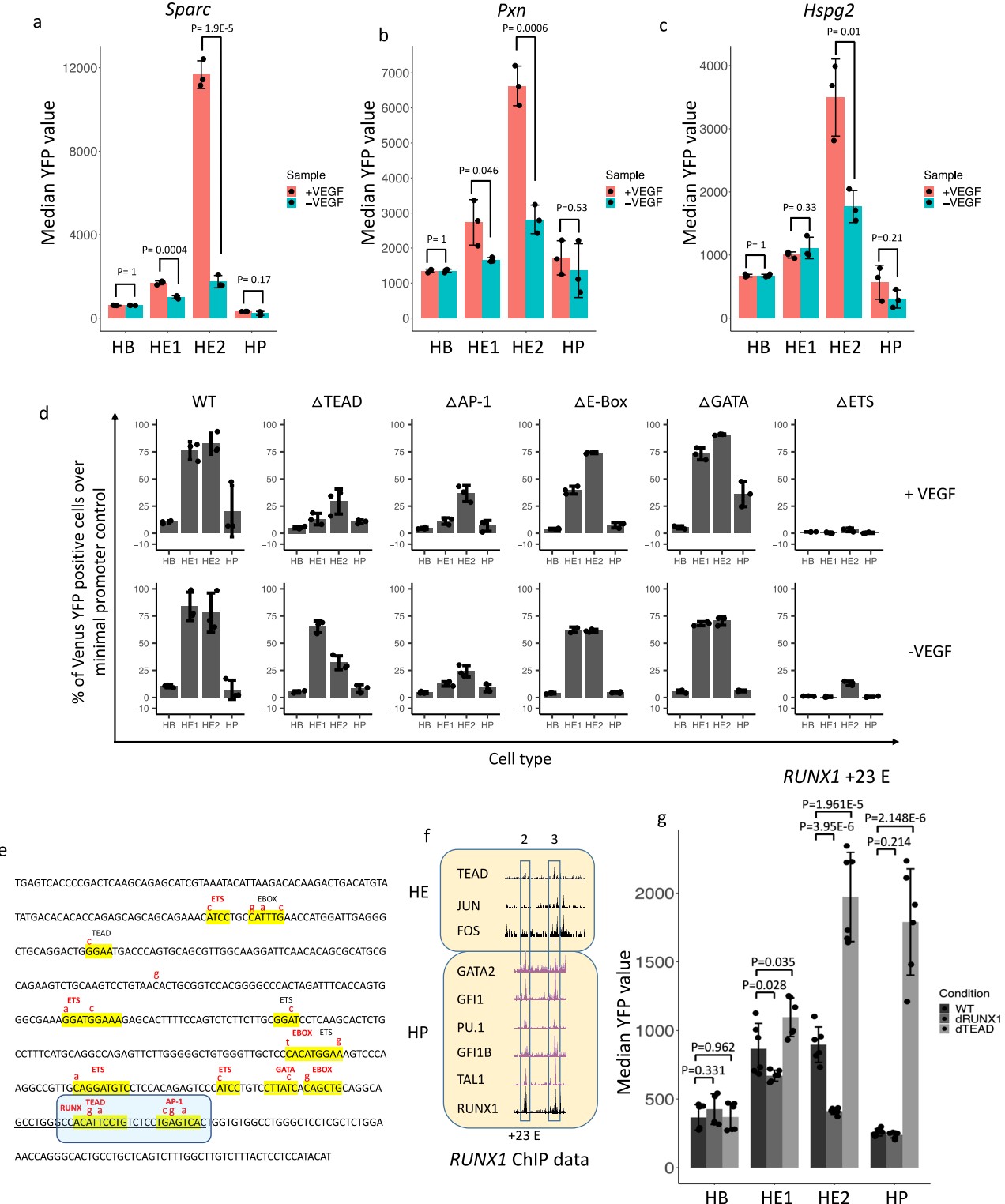

**Fig. 7 | HE-specific and cytokine responsive gene expression is mediated by the interplay of TEAD, AP-1 and RUNX1. a–c** Activity of the *Sparc, Pxn* and *Hspg2* enhancers (for details see Supplementary Notes) in isolation in the presence and absence of VEGF. Data are presented as mean values +/−SD. Dots showing individual values for *n* = 3 biologically independent experiments. *P*-values calculated using a two-sided Student's *t*-test. Source data are provided as a Source Data file. **d** HE-specific expression of *Galnt1* (data from ref. [18]), reporter gene activity driven by wild type (WT) and mutated *Galnt1* enhancer elements in the presence and absence of the indicated cytokines. Data are presented as mean values +/−SD.

Dots showing individual values for *n* = 3 biologically independent experiments. **e** Sequence of the *Galnt1* enhancer with TF binding motif mutations indicated in red. Strong consensus sequences are highlighted in red. Source data are provided as a Source Data file. **f** Chip data showing the binding of the indicated transcription factors to *RUNX1* enhancers in HE and HP cells. **g** Effect of mutations of the TEAD and RUNX1 binding sites on the individual activity of the *RUNX1* + 23 kb enhancer in serum culture. Data are presented as mean values +/−SD. Dots showing individual values for *n* = 6 biologically independent experiments. *P*-values calculated using a two-sided Student's *t*-test. Source data are provided as a Source Data file.

We previously identified HIPPO signalling as being crucial for HE development[18] and others have shown that shear stress activating the TEAD partner YAP via RHO-GTP induces the formation of HSCs from the HE via activation of *Runx1*[54]. Our ChIP data show that AP-1 and TEAD bind to such motifs in the HE[20] but once HP cells have formed, TEAD binding is lost[18]. We previously reported that TEAD and AP-1 often show a specific spacing between binding sites where binding is interdependent[20]. The expression of a dominant negative (dn)FOS peptide blocking all AP-1 DNA binding reduced the binding of TEAD at such sites, suggesting that HIPPO and MAP kinase signalling interface at these elements. In HP cells, RUNX1 has been shown to repress the endothelial program[48,53], we were therefore interested in understanding the role of RUNX1 in regulating enhancer activity. To examine the role of these TFs in more detail, we identified enhancer elements which were (i) active specifically in the HE, (ii) were bound by several of the above-described factors and (iii) were VEGF responsive (based on Supplementary dataset 4). We created ES cell lines carrying such enhancer sequences and examined their activity in serum-free cultures with and without VEGF. The *Galnt1* enhancer fitted these criteria (Fig. 7d, e, Supplementary Fig. 7a–c). It contains a composite AP-1/TEAD motifs (highlighted in blue) which overlaps with a RUNX1 binding site, together with a number of other motifs (Fig. 7e) allowing us to study the interplay between these factors in response to VEGF. Our ChIP data confirmed their binding (Supplementary Fig. 7d). dnFOS induction reduced *Galnt1* mRNA expression and TEAD binding specifically in the HE (Supplementary Fig. 7a, b), indicating that AP-1 activates the element and is required for the binding of TEAD factors within this composite module. We therefore decided to study the role of these factors in more detail by mutating the different binding sites.

To examine the response of the *Galnt1* enhancer to VEGF at four differentiation stages (HB – HP), we first measured the activity of the intact enhancer element (Fig. 7d) as compared to the promoter control by using flow cytometry to assay the number of YFP positive cells. The activity profile of the enhancer mirrored the gene expression profile in the presence of all cytokines (Fig. 7d). We also measured median YFP florescence confirming the results (Supplementary Fig. 7e). The analysis of mutant enhancer elements revealed that the binding motifs for TEAD, ETS and AP-1 were the most important for endothelial-specific median reporter activity compared to the wild-type sequence activity in the HE whereas the mutation of others had no impact (Fig. 7d, Supplementary Fig. 7e). The mutation of the TEAD and the overlapping TEAD/RUNX1 binding sites led to an increase in the percentage of YFP + cells in HE1 cells, suggesting that here TEAD and RUNX1 are in balance to restrict enhancer activity. Analysis of the TFs binding to the *Galnt1* enhancer revealed that in HP cells it is bound by RUNX1 and GFI1 (Supplementary Fig. 7d)[18] at a site overlapping the TEAD motif, suggesting a factor exchange and the establishment of a repressive RUNX1 complex. Taken together, these data demonstrate that VEGF modulates enhancer activity by the balanced interplay of RUNX1, AP-1 and TEAD TFs.

We next used the information obtained above to gain insight into the role of HE-and HP-specific TFs with regards to *RUNX1* regulation. To this end, we constructed ES lines harbouring mutations of the TEAD and RUNX1 binding sites in the + 23 kb enhancer (for ChIP data see Fig. 7e) to test how mutation affected developmental regulation in serum culture (for sequence details see Supplementary Notes, Fig. 7a). The + 23 kb element is activated in the HE and is then down-regulated (Fig. 7f). Consistent with what is seen with the *Galnt1* enhancer, mutation of the TEAD-site in the + 23 kb element led to an upregulation of the element in the HE, consistent with the repressive activity of this factor in the HE whilst mutating the autoregulatory RUNX1 site caused a reduction in enhancer activity.

Finally, we examined how VEGF impacted on other signalling factors involved in regulating HP development. The formation of the arterial HE as a source of HP cells is dependent on NOTCH1 signalling[46,55]. VEGF and SOX17 activate *Notch1* and lack of its down-regulation blocks the EHT[50,51,56,57]. *Notch1* and *Sox17* are down-regulated after the EHT and after upregulation in HP cells, RUNX1 and GFI1 repress *Sox17*[18,48,52,53] (Supplementary Notes, Fig. 2), thus forming a feed forward loop driving EHT progression. We examined whether any of the components of this pathway was modulated by VEGF. We found that the *Dlk1* gene encoding a repressor of NOTCH1 activity[58] was strongly upregulated in the absence of VEGF, both at the chromatin, enhancer activity and the gene expression level (Fig. 8a–c, Supplementary Notes, Fig. 4) as seen by sc-RNA-seq differential gene expression analysis in HE1 thus contributing to the regulation of the switch from an endothelial to a hematopoietic program. *Dlk1* is associated with two enhancers, one of which (enhancer 1) is VEGF-responsive and carries a TEAD / GATA / RUNX1 / E-Box signature (Supplementary Notes Fig. 8a, b).

Taken together, our data show that VEGF signalling in the hemogenic endothelium impacts on a gene regulatory network (Fig. 8d–f) that controls NOTCH1 signalling responsive enhancer elements which bind TEAD and AP-1 and are enriched in NOTCH signature (RBPJ/HES) and SOX motifs. The signalling-responsive activation pattern of these enhancers represents the molecular engine driving the EHT.

## Discussion

Our study introduces a versatile method that identifies developmental stage-specific, functional enhancers for any cell type that can be differentiated from ES cells. Importantly, our method recovers many enhancers that have been previously characterised in the context of whole loci, for example those in the Spi1 or the *RUNX1* locus[25,49]. The enhancer collection described here thus comprises an important resource for gene targeting approaches and for studies of how signals modulate enhancer activity. The method can also easily be adapted to human cells.

All identified cis-elements tested in our assay are derived from DNA-sequences existing as open chromatin and are enriched for motifs of differentiation stage-specific TFs known to be expressed at this stage. Integration with ChIP-Seq data from multiple sources confirms that these factors bind these motifs. The proportion of non-scoring fragments outside of a chromatin region scoring positive was surprisingly small, demonstrating that most open chromatin regions surrounding genes can impact on transcription. Inactive fragments most likely represent enhancer sub-fragments providing information about enhancer sub-structure. Distal elements identified in this study only partially overlap with chromatin features that have been associated with enhancer sequences, such as H3K27Ac. However, enhancers are nucleosome-free and are only flanked by modified histones. Nucleosomes cover 200 bp of DNA, each increasing the sequence space in which a functional element could be located. We therefore believe that assaying the function and sequence composition of open chromatin regions as described here will provide a much more precise tool to home in on those sequences with true transcription-regulatory activity. We see an association of distal elements with sites of RNA-Polymerase binding, confirming that enhancer transcription is widespread, but not ubiquitous.

We find that thousands of regulatory elements are connected to extracellular signals and contain binding motifs for signalling responsive TFs. In response to VEGF, thousands of chromatin regions are opened or closed, directly impacting on the expression of their associated genes. Our single cell experiments show that VEGF truly impedes differentiation, with *Runx1* failing to be upregulated, *Notch1* and *Sox17* not being downregulated, the EHT being delayed, and fewer HP cells being formed, thus directly linking signalling dependent enhancer activity to cell fate decisions. Based on this data, we suggest that the activity of most enhancer elements is fine-tuned by signalling which is therefore truly instructive. Our data comparing alternate

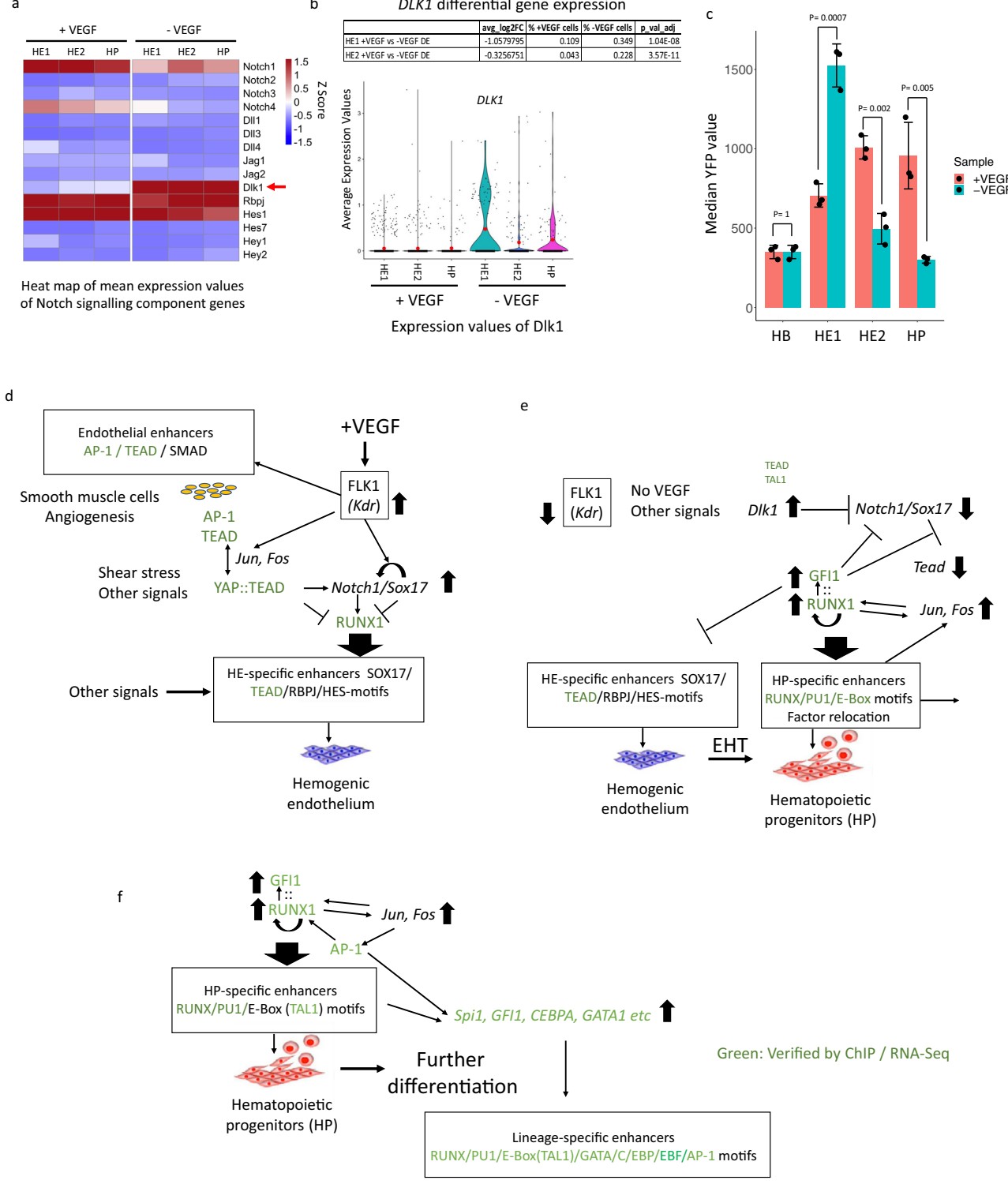

**Fig. 8 | VEGF regulates the balance between endothelial and hemogenic development by controlling NOTCH1 activity. a** Heatmap showing the expression of NOTCH signaling genes with (+VEGF) and without (−VEGF) based on single cell expression data. **b** Differential gene expression analysis of *Dlk1* depicted as Violin plots as described in Fig. 7 from cells cultured with and without VEGF across 6 clusters for *n* = 2266 cells (black line representing the median and red dot the mean). Differential gene expression taken as +/−0.25 average log2 fold change and

adjusted *P*-value < 0.01 (Bonferroni corrected). **c** The *Dlk1* enhancer 1 analyzed in isolation (Chr12:109,437,601-109438085) is VEGF responsive. Data are presented as mean values +/−SD. Dots showing individual values for *n* = 3 biologically independent experiments. *P*-values calculated using a two-sided Student's *t*-test. Source data are provided as a Source Data file. **d**–**f** Core gene regulatory and signaling network regulating blood stem cell emergence. For further details see text.

differentiation conditions show that signalling impact is highly culture dependent, which will make it imperative to seek differentiation conditions mimicking those found in vivo. In this context it is relevant that AP-1 motifs are enriched in all stage-specific enhancers and co-localize with multiple tissue-specifically expressed TF motifs, with the AP-1:TEAD / RUNX1 site in the *Galnt1* enhancer being an example. AP-1 can interact with chromatin remodellers to assist in the binding of other TFs[59], and multiple studies showed that it is involved in cell fate decisions in response to signals. For example, AP-1 binding is essential for signalling dependent chromatin priming and enhancer activation during T cell differentiation[60]. Blocking its activity at the HP stage during ES cell differentiation leads to a complete abolition of myelopoiesis[20]. The AP-1 family therefore represents a ubiquitous axis integrating the genomic response to specific external signals driving differentiation with pre-existing internal gene expression programs.

In addition to basic insights into enhancer function, our study provided additional mechanistic insights into how the core gene regulatory and signalling network regulating hematopoietic specification is connected to the genome (Fig. 8d–f). Via its receptor, FLK1, VEGF orchestrates differential enhancer and promoter activity which regulates the balance between the NOTCH1/SOX17 axis operating at specific enhancers establishing HE identity and RUNX1 driving the EHT and hematopoietic development. VEGF signals to AP-1[61,62], and we previously showed that this TF regulates the balance between hemogenic and vascular smooth muscle, i.e the endothelial fate[20]. AP-1 also interfaces with HIPPO signalling via TEAD TFs. Switching off HIPPO signalling via YAP activation induces *Runx1* expression in response to shear stress[54] and in endothelial cells promotes angiogenesis and endothelial gene expression[63]. During the EHT, RUNX1 is upregulated, autoregulates itself by binding to its own enhancers, represses the endothelial program by interacting with GFI1/LSD1 and relocates hematopoietic transcription factors such as TAL1/SCL and FLI-1 to hematopoietic genes[52] leading to further differentiation (Fig. 8e). Taken together with these findings, our ChIP, single-cell gene expression and reporter gene studies place the interplay between RUNX1, SOX17, NOTCH, HIPPO-signalling and AP-1 mediating VEGF responsive TFs at specific enhancer elements at the heart of the balance between endothelial or hematopoietic fate. Although most of the network components and their different roles are known from many perturbation and knock-out experiments as reviewed in[44], it was unclear how and where signals and transcription factors impact on the genome and how they are connected within cell stage specific gene regulatory networks. In this study, we have now identified the factors involved in their regulation and the genomic elements upon which they act, providing a rich resource for studies of identifying the signals required for the activation of the correct gene expression program required for efficient blood cell production.

## Methods

### HM-1 ES cell culture

The HM-1 targeting ES cell line originally described by Magin et al (1992)[64] was cultured on gelatinised tissue culture plates in DMEM-ES media (DMEM (Merck, D5796) with 15% FSC, 1 mM sodium pyruvate (Merck, S8636), 1 x Penicillin/Streptomycin (Merck, P4333), 1 x L-Glutamine (Merck, G7513), 1 x Non-Essential amino acids (Merck, M7145), 1000 U/ml ESGRO®LIF (Merck, ESG1107), 0.15 mM MTG and 25 mM Hepes buffer (Merck, H0887)) at 37 °C and 5% $CO_2$. Cell colonies were dissociated with TrypLE™ Express (ThermoFisher, 12605010) at 48 h intervals and re-plated at $1.2 \times 10^4$ per cm².

### In vitro hematopoietic differentiation in Serum

Hematopoietic In vitro differentiation (I.V.D) in serum was essentially performed as previously described in Obier et al.[20]. ES cells colonies were dissociated to form a single cell suspension using TrypLE™ Express (ThermoFisher, 12605010) and trypsin activity was stopped by

addition of DMEM-ES media at 1:1 ratio. Cells were then resuspended at $2.5 \times 10^4$/ml in I.V.D media (IMDM (Merck, I3390), 15% FCS, 1 x Penicillin/Streptomycin (Merck, P4333), 1 x L-Glutamine (Merck, G7513), 0.15 mM MTG, 50 µg/ml Ascorbic acid and 180 µg/ml Human transferrin (Merck, T8158)) and plated onto non-adherent dishes (Thermo-Fisher, 501 V). The cells were the incubated at 37 °C, 5% $CO_2$ for 3 days until floating embryoid bodies (EBs) were formed.

Embryoid bodies were harvested by transferring EB containing media to 50 ml centrifuge tubes and allowing the EBs to settle by gravity. The EBs were washed with PBS, allowed to settle and then were dissociated with TrypLE™ Express (ThermoFisher, 12605010) ensuring a single cell suspension was achieved while avoiding damage to the cells. The dissociated cells were then sorted for the FLK1 surface marker (forming the population referred to as HB), by incubation with a FLK1 biotin-coupled antibody (1:200) (eBioscience, 13-5821-82), followed by mixing cells with MACS anti-biotin beads (Miltenyi Biotec, 130-090,485) and separation using a MACS LS column (Miltenyi Biotec, 130-042-401). The sorted cells were then used for flow cytometry and sequencing experiments as well as further differentiation in blast culture into HE1, HE2 and HP.

Blast culture was performed by resuspending the FLK1+ cells and plating on a gelintin coated tissue culture flask at $1.6 \times 10^4$ cells per cm² in Blast media (IMDM (Merck, I3390), 10% FCS, 1 x Penicillin/Streptomycin (Merck, P4333), 1 x L-Glutamine (Merck, G7513), 0.45 mM MTG, 25 µg/ml Ascorbic acid, 180 µg/ml Human transferrin (Merck, T8158), 20% D4T Conditioned Media, 5 ng/ml VEGF(PeproTech, 450-32), 10 ng/ml IL-6 (PeproTech, 216-16)) and incubating for 2.5 days at 37 °C, 5% $CO_2$.

FACS sorting of the HE1, HE2 and HP populations was achieved after incubation with antibodies KIT-APC (1:100)(BD Pharmingen, 553356), TIE2-PE (1:200)(eBioscience, 12-5987-82) and CD41-PECY7 (1:100)(eBioscience, 25-0411-82) and then sorted into HE1 (KIT+, TIE2+, CD41-), HE2 (KIT+, TIE2+, CD41+) and HP (KIT+, TIE2-, CD41 +) populations.

### Serum free in vitro hematopoietic differentiation

Serum Free I.V.D Culture derived Embryoid bodies (EBs) were generated from HM-1 mouse embryonic stem cells by plating at $5.0 \times 10^5$ cells/ml in StemPro™−34 SFM media (ThermoFisher, 10639011) into non-adhesive dishes (ThermoFisher, 501 V). BMP4 (PeproTech, 315-27) was added at a concentration of 5 ng/ml. Cultures were left to incubate at 37 °C and 5% $CO_2$ for 60 h before bFGF (PeproTech, 450-33) and Activin A (PeproTech, 120-14E) were added at a concentration of 5 ng/ml each. The cells were incubated for 16 h at 37 °C, 5% CO2 and then sorted for FLK1 + cells as described in Obier et al.[20]. FLK1 + cells were plated in StemPro™−34 SFM media on 0.1% gelatine coated plates or tissue culture flasks at $2.25 \times 10^4$ cells per cm². BMP4, Activin-A and bFGF were added to a concentration of 5 ng/ml for 16 h. Media was then removed, the blast culture was washed with PBS (Merck, D8662) and fresh StemPro™−34 SFM media was added containing BMP4 (5 ng/ml), VEGF (5 ng/ml)(PeproTech, 450-32), TPO (5 ng/ml))(PeproTech, 315-14), SCF (100 ng/ml))(PeproTech, 250-03), IL6 (10 ng/ml)) (PeproTech, 216-16) and IL3 (1 ng/ml))(PeproTech, 213-13). For cytokine withdrawal experiments, one of BMP4, VEGF, IL6, or IL3 was not added at this stage. Blast cultures were left to incubate at 37 °C and 5% $CO_2$ for 72 h before cells were harvested for cell sorting and FACS analysis. Inhibition of trypsin was achieved using Trypsin Inhibitor (ThermoFisher, 17075029) at a 1:1 ratio.

### Enhancer reporter library cloning

A genome-wide enhancer reporter assay was designed based on an enhancer reporter system designed by Wilkinson et al. (2013)[23]. Briefly the enhancer reporter functions by inserting a fragment of interest upstream of a HSP68 minimal promoter and Venus-YFP reporter by Gateway® cloning. The reporter construct is then transfected into the

HM-1 ES cell line which has a non-functional HPRT locus. Using HPRT homology arms the report cassette becomes integrated into the HPRT locus by homologous recombination, also repairing the locus, enabling selection of clones with successful recombination. This enhancer reporter system was modified and optimised for genome-wide screening. To obtain genome-wide enhancer fragments for cloning into the reporter we isolated tn5 tagmented open-chromatin fragments based on the ATAC-Seq protocol[33] from cells of the five differentiation stages. Cells were obtained ($2.5 \times 10^5$) from each differentiation stage (ES, HB, HE1, HE2, HP) as detailed in the serum IVD method. Cells were pelleted in 5 aliquots of $5 \times 10^4$ cells and following the ATAC-Seq protocol each cell pellet had a transposition mix added (2x TD Buffer 25 µl, TN5 Transposase 2.5 µl (Illumina, 20034197), PBS 16.5 µl, 1% Digitonin 0.5 µl (ThermoFisher, BN2006), 10 % Tween-20 0.5 µl (Merck, P9416), $H_2O$ 5 µl) and cells were gently resuspended by pipetting. The transposition reaction was then incubated in a shaker at 700 RPM, 37 °C for 30 min. The 5 reactions were then combined and DNA purified using the Qiagen MinElute® Reaction Clean-up kit (28206) and eluted in 26.5 µl $H_2O$. One fifth of the purified reaction from each stage was used to produce the baseline ATAC-Seq libraries (see ATAC-Seq method) and the remainder had linkers added incorporating AttB Gateway® cloning sites to enable insertion into a Gateway® donor vector (pDONR 221, Thermo Fisher). The linkers were designed based on the tn5 transposase sequence and added by PCR in 2 steps. Initially 5 cycles of PCR were performed (Transposed DNA 20 µl, $H_2O$ 20 µl, 25 µM AttB tn5 Fwd primer 5 µl, 25 µM AttB tn5 Rev primer 5 µl, NEBNext Master Mix 50 µl (NEB, M0541)) with the following conditions, 72 °C for 5 min, 98 °C for 30 s, 5 cycles of 98 °C 10 s, 63 °C 30 s and 72 °C 1 min and a final hold at 4 °C.

To optimize the required cycle number, the material from the initial PCR reaction was split and a 5 µl was used for a separate qPCR reaction (0.125 µM AttB tn5 Fwd Primer, 0.125 µM AttB tn5 Rev Primer, 1 x SYBR Green I (Merck, S9430), NEBNext Master Mix (NEB, M0541)) with the following conditions: 72 °C for 5 min, 98 °C for 30 s, 35 cycles of 98 °C 10 s, 63 °C 30 s and 72 °C 1 min using a StepOnePlus™ Real-Time PCR System (Thermo Fisher). Using the raw data from the cyan channel the number of cycles for 25% amplification was calculated. The number of cycles obtained from the qPCR side reaction was then used to amplify the remainder of the material as follows, 98 °C for 30 s, cycles of 98 °C 10 s, 63 °C 30 s and 72 °C 1 min with a final hold at 4 °C. The resulting material was size selected, to optimise for material originating from enhancers, for fragments between 350 and 650 bp by electrophoresis on a 1.2% agarose gel with ethidium bromide and gel extraction was performed using the Qiagen MinElute® Gel Extraction Kit (Qiagen, 28604) following manufactures protocol.

To generate attL-flanked entry clones compatible with cloning into the reporter vector the size selected material was cloned into the pDONR™221 (Thermo Fisher) by BP Gateway® reaction (100 fmol pDONR221, 100 fmol PCR material, 2 µl BP Clonase II, made up to 20 µl with TE buffer (pH 8.0) the reaction was incubated for 18 h at 25 °C and ended by treatment with 0.2 µg proteinase K and incubation at 37 °C for 10 min. NEB 10-beta Electrocompetent cells (NEB, C3019H) were transformed with the plasmid by electroporation with a Bio-Rad GenePulser using program EC1 (2.0 kV, 200 Ohm, 25 µF), after addition of outgrowth media and incubation at 37 °C, 250 rpm for 1 h the bacteria were spread onto 150 mm agar plates containing 50 µg/ml kanamycin. Following overnight incubation at 37 °C, 10 ml of LB was added to each plate and the bacterial lawn scraped off using a cell scraper. Plasmid DNA was extracted from the collected bacteria by Maxi-prep using the NucleoBond Xtra Midi EF kit (Macherey-Nagel, 740420.50) following manufacturer's instructions. The resulting product was eluted in 500 µl $H_2O$.

The chromatin fragments were then transferred into the enhancer reporter cassette in the pSKB-GW-Hsp68-Venus vector[23] by Gateway® LR reaction (150 ng Entry clone (pDONR), 150 ng pSKB, 2 µl LR Clonase II, made up to 10 µl with TE buffer (pH 8.0)) the reaction was incubated for 1 h at 25 °C and ended by treatment with 0.2 µg proteinase K and incubation at 37 °C for 10 min. As with the BP reaction product, NEB 10-beta Electrocompetent cells were transformed with the material and after overnight incubation on agar plates containing 200 µg/ml ampicillin the bacteria were collected and the plasmid library extracted by Maxi-Prep. To obtain the highest possible diversity of fragments the cloning was carried out 10 times for material from each differentiation stage with the pDONR 221 library material being combined before moving to the LR clonase reaction.

### Enhancer reporter library transfection

The HM-1 ES cell line was cultured on gelatinised culture plates in ES-Media (see IVD method) at 37 °C, 10% $CO_2$ and split every 48 h to maintain healthy cells. Prior to transfection the cells were treated for 1 week with $1 \times 6$-TG (6-Thioguanine) (Merck, A4660) to remove any cells with a functional HPRT locus. The plasmid libraries were mixed and digested with PmeI (NEB, R0560) to linearise the plasmids prior to transfection. To obtain optimum homologous recombination efficiency CRISPR guides were designed flanking the region and cloned into the PX458 Cas9 and sgRNA expression vector[65] (pSpCas9(BB)−2A-GFP (PX458) was a gift from Feng Zhang (Addgene plasmid # 48138; http://n2t.net/addgene:48138; RRID:Addgene_48138).

For each experiment, $500 \times 10^6$ cells were transfected using a Nucleofector®−4D (Lonza) with the P3 Primary Cell X kit (Lonza, V4XP-3024) with $5 \times 10^6$ per cuvette 10 ug pSKB enhancer reporter plasmid and 10 ug of PX458-HPRT CRISPR plasmid using program CG-104. Following electroporation, the cells were plated on gelatinised plates in ES-media with the addition of 50 µM SCR7 pyrazine (Merck, SML1546) to inhibit Non-homologous end joining, promoting homologous recombination following cutting by Cas9. After 12 h the media was changed and 1 x HAT (Hypoxanthine-Aminopterin-Thymidine (Merck, H0262)) was added to select for clones with successful recombination. The cells were then maintained in media containing HAT for one week and HT (hypoxanthine and thymidine (Merck, H0137)) for a further 2 passages to prevent the cells from dying after the withdrawal of HAT, whereby all cells were kept and replated after splitting.

The selected cells were then put into IVD cultures and differentiated into HB, HE1, HE2, and HP and sorted by FACS (See Serum IVD Method). In addition to the previously mentioned gating for obtaining the cell populations the cells were also sorted into negative, low, medium and high Venus-YFP populations representing the activity of the enhancer fragment driven minimal promoter. The YFP-Venus negative population was gated based on a cell line containing a reporter construct with only the minimal promoter and reporter with no enhancer fragment.

### Enhancer reporter library preparation

Genomic DNA was extracted from the sorted cell populations using the Qiagen DNA Micro kit following manufacturer's instructions. To amplify the fragments contained in the reporter cassettes and add sequencing adaptors a PCR using indexed primers compatible with Illumina sequencing and targeted against the tn5 transposase sequence ((0.5 µM Nextera PCR Primer i5, 0.5 µM Nextera PCR Primer i7, 25 µl NEBNext Master Mix). Dependent on the amount of DNA obtained from the sorted cells between 25 and 30 PCR cycles were performed using the following conditions 98 °C for 30 s, cycles of 98 °C 10 s, 63 °C 30 s and 72 °C 1 min with a final hold at 4 °C. Libraries were quantified using a Bioanalyzer 2100 (Agilent) with a High

Sensitivity DNA Chip and pooled at equal concentration for sequencing. The final pool concentration and size was then confirmed by Kappa Library Quantification Kit (Roche, KK4824) and Bioanalyzer. The pooled libraries were sequenced on a Next-Seq 550 as a paired end run with a 150 cycle High Output kit by the Genomics Birmingham Sequencing Facility.

**Trouble-shooting note.** During the development of the method, we discovered by sequencing that the different cloning and targeting vectors reproducibly preserved the original complexity of the ATAC-Seq library. The number of individual sequences equalled the sequence complexity contained in the original ATAC-Seq experiment. We recommend performing such controls for every experiment. However, as shown in the manuscript, the screening procedure can give different levels of enhancer coverage. It turned out that the quality and viability of the transfected cells and their differentiation capacity, i.e., the ability of each individual clone carrying one construct to form the different cell types, was crucial to obtain a maximal coverage of the initial complexity of the library. Therefore, care needs to be taken to maintain the cells in optimal growth condition and use a large number of cells as recommended above.

**Generation of cell lines carrying individual reporter constructs**
A putative enhancer for the *Galnt1* gene containing multiple transcription factor binding sites was identified from DNaseI hypersensitive site data[18]. To test the functionality of the enhancer throughout differentiation the same enhancer reporter system[23] as used for the genome-wide screen was employed. The genomic sequence for the enhancer was obtained and flanking attB1 and attB2 sites were added to enable Gateway® cloning (Supplementary Notes Table 3). This fragment was then synthesised as an Invitrogen GeneArt Strings DNA Fragment (ThermoFisher). The resulting fragment was cloned by the one step Gateway protocol whereby in a single reaction the fragment was inserted into pDONR 221 to generate an attL-flanked entry clone by BP clonase II (ThermoFisher, 11789020) and from this vector into the pSKB GW-Hsp68-Venus reporter vector[23] by LR clonase II (Thermo-Fisher, 11791020). Briefly this reaction consisted of 100 ng of Enhancer DNA, 75 ng pDONR 221, 75 ng pSKB GW-Hsp68-Venus, TE pH 8.0 to 6 µl total volume, 1.5 µl LR clonase and 0.5 µl BP clonase. The reaction was incubated at 25 °C for 3 h and then had 0.2 µg proteinase K added and a further incubation for 10 min at 37 °C to stop the reaction. Competent DH5α bacteria (NEB, C2987H) were transformed with 1 µl of Gateway® reaction by incubation on ice for 30 min and heat shock at 42 °C for 45 s. The bacteria were then incubated for 1 h at 37 °C with SOC media (ThrmoFisher, 15544034) and plated onto agar plates containing 100 µg/ml ampicillin (Merck, A9518). After overnight incubation at 37 °C colonies were picked for mini-prep cultures and mini-preps were performed using the Qiaprep® Miniprep kit (Qiagen, 27106 × 4). The insert in the resulting plasmid was sanger sequenced (Source Biosciences, stock sequencing primer 'EGFP_Nrev') to check the sequence. Following confirmation of the insert sequence the same bacterial colony was used to grow cultures for Maxi-prep which was performed using the EndoFree® Plasmid Maxi Kit (Qiagen, 12362).

HM-1 cells were cultured and transformed in the same way as with the Enhancer screen method. Following 6-TG treatment $5 \times 10^6$ HM-1 cells were transfected using a Nucleofector®–4D (Lonza) with the P3 Primary Cell X kit (Lonza, V4XP-3024). The cells were then selected for those with successful integration of the reporter cassette by treatment for 5 days with 1 x HAT containing media. At 9 to 12 days after colonies appeared clones were picked and re-plated on a gelatinised 96 well plate. The clones were then grown in media containing 1 x HT for 2 passages and then used for IVD and flow cytometry as detailed.

To study the impact of different transcription factor binding on the *Galnt1* enhancer, sequences were produced where various transcription factor binding motifs were mutated. These mutant versions

of the enhancer were cloned in the same way as with the original *Galnt1* enhancer sequence (Supplementary Notes Table 3).

To validate enhancer elements identified in our enhancer, enhancers associated to the *Sparc, Pxn, Eif2b3, Dlk1* and *Hspg2* genes were chosen. These enhancers were also identified as being VEGF signalling responsive taken as a 2-fold change in ATAC-seq peak height with and without VEGF in the culture. Primers (Supplementary Notes Table 4) were designed to amplify out the 400 bp enhancer element and then the enhancer element was cloned HM-1 mESCs as described above.

To validate the + 23 kb and + 3.7 kb *RUNX1* enhancers, the 400 bp enhancer sequences were synthesised by GeneArt String synthesis (Thermo Fisher). For each *RUNX1* enhancer with individual TF binding motifs mutated GeneArt String synthesis (Thermo Fisher) was used (Supplementary Notes Table 3). The sequences were then cloned into HM-1 mESCs as described above.

**ATAC-sequencing.** ATAC-Seq was performed as described in Buenrostro et al.[66]. Briefly 5000-50,000 HB, HE1, HE2 and HP cells were sorted by FACS and transposed in 1x tagment DNA buffer, Tn5 transposase (Illumina, 20034197) and 0.01% Digitonin (ThermoFisher, BN2006) for 30 min incubated at 37 °C with agitation. DNA was purified using a MinElute Reaction Cleanup Kit (Qiagen) and DNA was amplified by PCR using Nextera primers[66].

**Single cell sorting and single cell RNA-Seq.** $7.0 \times 10^6$ cells from the All cytokines condition blast culture and $2.8 \times 10^6$ cells from the VEFG withdrawal blast culture were taken for cell sorting and sc-RNA-Seq library preparation. Both samples were stained with CD41-PECY7 (1:100)(eBioscience, 25-0411-82), KIT-APC (1:100)(BD Pharmingen, 553356) and TIE2-PE (1:200)(eBioscience, 12-5987-81) and then sorted into HE1 (KIT + , TIE2 + , CD41-) and HE2 (KIT + , TIE2 + , CD41 + ) populations. From the All cytokines sample, 758000 HE1 and 380000 HE2 cells were sorted. From the VEGF withdrawal sample 71500 HE1 and 33300 HE2 cells were purified by cell sorting. Cells were resuspended in 80 µL at a concentration of 1000–1200 cells/µL for evaluation of cell viability. The viability of the All cytokines populations as measured by Trypan Blue (Merck, T8154) staining was found to be 73% for HE1 and 92% for HE2. The viability of the VEGF withdrawal populations was found to be 73% and 75%. For each sample, 10,000 single cells were loaded on a Chromium Single Cell Instrument (10x Genomics) and processed.

**Data analysis methods**
**Enhancer sequencing data analysis.** Paired end sequencing reads were trimmed using Trim Galore (https://github.com/FelixKrueger/TrimGalore)[67] with the parameters --nextera --length 70 –paired to remove remaining adaptor sequences and poor-,quality sequences. The reads were then aligned to the mm10 genome using bowtie2[68] using parameters --very-sensitive --fr --no-discordant -X 600 --no-mixed, a maximum fragment length (-X) of 600 bp was set based on the largest size selected fragments during the initial cloning. Aligned reads were filtered for only those which were properly paired and with a mapq score of over 40 using Samtools and output as a bedpe file. Because we treat each unique read as a direct readout of enhancer activity, this filtering is an essential step to remove multimapping reads which would produce false positives. The bedpe file was further converted to a bed file of enhancer fragments by taking the first co-ordinate of read 1 and the final co-ordinate of read 2 for each pair. Duplicate fragments were removed using the unique function leaving a bed file of unique aligned fragments representing those PCR amplified from the FACS sorted cells.

Because we use the fragments generating a positive FACS signal as a direct readout of enhancer activity stringent filtering is essential.

To remove potential PCR artefacts, a library was produced using the same PCR conditions as the other libraries but using DNA from untransfected cells. Since there should be no complimentary sequences to the ATAC primers in wild type DNA any fragments produced in this background library represent PCR artefacts. Using the background library, fragments were further filtered using the intersect function on Bedtools[69] and any having a perfect overlap with the background library were removed. The fragments were then filtered using Bedtools intersect for those which were found in open chromatin, defined by DNaseI-Seq, in Goode et al. (2016)[18] and further by a corresponding ATAC peak in the correct differentiation stage in HM-1 cells. This enabled us to define which stage a reporter fragment would be in open chromatin and have potential to act as an enhancer. Finally, the enhancer fragments were annotated using the annotatePeaks function of Homer for those within 1.5 kb of a TSS (promoter fragments) and distal fragments representing genuine enhancers. This analysis produced a final list of enhancer fragments which was used for all further fragment analyses. We defined full enhancers by overlapping distal ATAC sites with enhancer fragments and called any open chromatin region with an overlapping positive enhancer fragment as enhancer positive. The remaining ATAC sites we define as negative/unknown because negative fragments may not overlap the required TF binding sites for enhancer activity.

**ATAC-seq data analysis.** Single-end reads from ATAC-seq experiments were processed with Trimmomatic (version 0.39) and aligned to the mm10 mouse genome using Bowtie2 2.4.4[68] using the options -very-sensitive-local. Open chromatin regions (peaks) were identified using MACS2 2.2.7.1[70] using the options -B -trackline -nomodel. The peak sets were then filtered against the mm10 blacklist[71]. Peaks were then annotated as either promoter-proximal if within 1.5 kb of a transcription start site and as a distal element if not. To conduct differential chromatin accessibility analysis, a peak union was first constructed by merging peaks from two comparisons. Tag-density in a 400 bp window as centred on the peak summits was derived from bedGraph files from MACS2 peak calling using the annotatePeaks.pl function in Homer 4.11[72] (with the options -size 400 -bedGraph. Tag-densities were then normalised as counts-per-million (CPM) in R v3.6.1 and further log2-transformed as $\log_2(\text{CPM}+1)$. In cases were replicate samples were available, the average normalized tag count was used for all downstream comparisons. A peak which was two-fold increase or decrease different to the control was taken as being differentially accessible. A de-novo motif analysis was performed on sets of gained and lost peaks using the find -MotifsGenome.pl function in Homer using the options -size 200 -noknown. Tag density plots were constructed by retrieving the tag-density in a 2 kb window centred on the peak summits with the annotatePeaks.pl function in Homer with the options -size 2000 -hist 10 -ghist -bedGraph. These were then plotted as a heat map using Java TreeView v1.1.6[73].

**Gene annotation using HiC data.** Promoter capture HiC data from ESCs cells were downloaded from Novo CL et al. (accession numbers: GSM2753058, runs: SRR5972842, SRR5972842, SRR5972842, SRR5972842, SRR5972842)[34] and from HPC7 from Comoglio et al.[74] (accession numbers: GSM2702943 & GSM2702944, runs SRR5826938, SRR5826939). The CHi-C paired-end sequencing reads were aligned to the mouse genome mm10 build using HiCUP pipeline[75]. Initially, the raw sequencing reads were separated and then mapped against the reference genome. The aligned reads were then filtered for experimental artefacts and duplicate reads, and then re-paired. Statistically significant interactions were called using GOTHiC package[76] and HOMER software[72]. This protocol uses a cumulative binomial test to detect interactions between distal genomic loci that have significantly more reads than expected by chance, by using a background model of random interactions. This analysis assigns each interaction with a p-value, which represents its significance. The union of all CHiC interactions from both ESC and HPC7 cells were used to annotate positive enhancer to their related promoter.

**Motif co-localization analysis.** Genomic co-ordinates for each transcription factor (TF) binding motif were retrieved from the sets of stage specific positive enhancer fragments and from within all distal ATAC-Seq peaks using the annotatePeaks.pl function in Homer and exported as a BED file using the -mbed option. Motif co-occurrence was then measured for each stage (ES, HB, HE1, HE2 and HP) by counting the number of times a pair of TF motifs were found within 50 bp of each other in the set of specific positive enhancer fragments.

To assess the significance of this co-occurrence, we carried out a re-sampling analysis whereby a number of ATAC-Seq sites equal to the number of stage specific enhancer fragments was randomly sampled from the set of all distal ATAC sites found in that stage. The number of motif pairs was then counted in this random set. This procedure was repeated 1000 times and resulted in a distribution of motif pair counts for each pair of TF binding motifs. A z-score was then calculated for each motif pair using as:

$$z = \frac{x - \mu}{\sigma} \tag{1}$$

Where x is the number of motif pairs found in the specific positive enhancer fragments, μ is the average number of motif pairs in 1000 random samples, and σ is the standard deviation for those motif pair counts. A positive z-score in this case suggests that the number of motif pairs found in the positive enhancer fragments is greater than could be expected by chance. The resulting z-score matrix was then hierarchically clustered using complete linkage of the Euclidean distance in R and displayed as a heatmap.

**Relative motif enrichment analysis from ATAC-Seq data.** To identify transcription factor binding motifs which were enriched in a set of peaks relative to another, we calculated a relative enrichment motif score, $S_{ij}$ for each motif i in each peak set j as:

$$S_{ij} = \frac{\frac{n_{ij}}{m_j}}{\sum_j \frac{n_{ij}}{\sum_j m_j}}, \tag{2}$$

where $n_{ij}$ is the number of instances of motif i in peak set j and $m_j$ is the total number of sites in peak set j. This score was calculated for each TF motif in each of the peak sets and a matrix of enrichment scores was produced which was then hierarchically clustered using complete linkage of the Euclidean distance in R and displayed as a heat map with results scaled by either row or column.

**Single cell RNA-Seq analysis.** Fastq files from scRNA-Seq experiments were aligned to the mouse genome (mm10) using the count function in CellRanger v6.0.1 from 10x Genomics and using gene models from Ensembl (release 102) as the reference transcriptome. The resulting Unique Molecular Identifier (UMI) count matrices were processed using the Seurat package v4.0.5[77] in R v4.1.2. Cell quality control was carried out for each of the four samples sequenced individually in order to remove cells which had few or a higher than expected number of detected genes or that had a high proportion of UMIs aligned to mitochondrial transcripts or quality control parameters for each sample. Genes detected in less than 3 cells were removed from the analysis.

The filtered data objects for each cell type were then combined according to their VEGF status and normalized using the Log-Normalize method. In order to calculate the cell cycle stage for each of the cells, the in-built S-phase and G2M-phase marker gene lists

from Seurat were first converted from human gene symbols to their corresponding mouse orthologs using the biomaRt package v2.50.0[78] in R. Cell cycle stage was then inferred using the CellCycleScoring function in Seurat. The possible effect of cell cycle stage on downstream analysis was then removed from the dataset by linear regression using the ScaleData function. Clustering of cells was performed using the first 20 principal components and visualized using the Uniform Manifold Approximation and Projection (UMAP) method. Cell marker genes for each of the clusters identified (Supplementary Dataset 5) were calculated using the FindAllMarkers function. A gene was considered a marker gene if it was expressed with a log2 fold-difference of 0.5 between the cluster being considered and all other cells as well as being detected in at least 50% of cells in that cluster. A gene was considered statistically significant if it had a Bonferroni-adjusted $p$-value < 0.05. Cell type was then inferred for each cluster identified by manual inspection of the marker gene lists and comparing these to the expression known surface markers genes (Tie2, Cd41) for HE1, HE2, and HP.

Cell trajectory (pseudotime) analysis was carried out using Monocle v3.1.0[79]. Data from Seurat was exported to Monocle using the seuratwrappers package in R (https://github.com/satijalab/seurat-wrappers). Trajectories were then inferred using the learn graph function and ordered along pseudotime by selecting a root node which corresponded to the earliest cell-type (HE1).

### Reporting summary

Further information on research design is available in the Nature Portfolio Reporting Summary linked to this article.

## Data availability

The data that support this study are available from the corresponding authors upon reasonable request. The genome-wide data generated in this study have been deposited in the Gene Expression Omnibus (GEO) database under accession code GSE198775. The genome-wide data generated in this study have also been provided as a UCSC Genome Browser Track-Hub (https://genome-trackhub.bham.ac.uk/data/EnhancerHub/hub.txt). The publicly available genome-wide data analysed in this study were deposited in GEO under accession code GSE69101, GSE143460, GSE126496 and GSE79320. Source data are provided with this paper.

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

## Acknowledgements

This research was funded by a project grant from the Biotechnology and Biological Sciences Research Council (BBSRC) to C.B. and J.B. (BB/R014809/1), a BBSRC MiDTP studentship to C.B. for A.M., a BBSRC MIBTP studentship to A.M., a BBSRC LoLa grant to C.B and B.G. (BB/I001220/2), as well as grants from Blood Cancer UK (15001) and the Medical Research Council (MR/S021469/1) to CB and P.N.C. We thank Genomics Birmingham for expert sequencing services, Nunzia Nirchio for technical support and the Birmingham Technology Hub for cell sorting facilities.

## Author contributions

B.E.-W., A.M., S.K., L.A., D.G., and M.C. performed experiments, generated and analysed data, P.K., S.A.A., and I.P. analysed data, J.B. supervised data analysis and together with P.N.C. analysed data and helped writing the manuscript, B.G. contributed resources and helped writing the manuscript, C.B. conceived and directed the study and C.B. and B.E.-W. wrote the manuscript.

## Competing interests

The authors declare no competing interests.
