## [Peer Review File · Nature Communications]

REVIEWER COMMENTS

Reviewer #1 (Remarks to the Author):

The manuscript by Edginton-White describes a genomic effort to discover cis-regulatory elements operational during hematopoietic differentiation of mouse embryonic stem cells. The authors indicate that the analysis yielded “thousands of differentially active cis-elements” at multiple stages of differentiation. A major conclusion highlighted was that stage-specific cis-elements respond to cytokine signals that impact activity of signal-dependent transcription factors. It was stated that the work “presents a major advance in our understanding of developmental gene expression”. The problem of how large numbers of cis-regulatory elements operate and are regulated during cellular transitions has obvious importance and is studied by many investigators. There have been many contributions to the problem, and the authors have selectively cited literature on this problem. There are important limitations of the current study.

1) While the overall approach utilised has potential to yield significant insights, the major conclusions recapitulate known concepts. Multiple groups have described signal-dependent (cytokine, stress etc.) enhancers controlled by signal-dependent transcription factors (e.g. Choudhuri,,,, Zon Nature Genetics and multiple prior examples). This is a common paradigm that has been studied at the gene-specific level and via genomic approaches. Signal-dependent transcription factors often co-localize with hematopoietic transcription factors at enhancers, illustrating how cellular signaling mechanisms impact genome function. Overall, it is not at all evident what the “major advance” is.

2) A second major limitation relates to the argument presented in the introduction, highlighting the limitations of reporter gene assays to define cis-regulatory elements and the challenges of discovering elements operational in chromatin. A key point that is not addressed is that years of studies have demonstrated the importance of analyzing cis-regulatory element function at endogenous loci and not in the context of transgenes integrated at ectopic sites. At the very least, discoveries utilising transgenes need to be validated at endogenous loci, and many prior studies have indicated that activity at ectopic loci does not predict endogenous locus activity – quantitatively and even qualitatively. Even with the early studies on the beta-globin locus, hypersensitive sites and cis-regulatory elements within them almost always were important at ectopic chromatin sites (e.g. linked to reporters, analogous to the author’s system), but careful analyses at endogenous loci demonstrated the promiscuity of elements at ectopic sites, which were often not essential at the endogenous locus. These studies have been extended well beyond the globin genes. Thus, to bolster arguments that this reporter-based strategy has unique utility for discovering important (or essential) cis-regulatory elements and that the study has yielded a valuable “resource”, data needs to be provided that tests cis-regulatory element function at the respective endogenous loci.

Additional Comments:

1) Methodological details – Fig. 1 legend and text in Results do not clearly describe the high-throughput strategy. There was little to no discussion of issues related to quantitation, statistical analyses, limitations of using the reporter system etc.

2) Fig. 2 – “overlapping”

3) Legends commonly lacked statistical strategies, and most figures lacked this essential analytical information.

4) Fig. 6 – The authors were interested in Galnt1. Galnt1 is a broadly expressed gene, with mRNA present in most, if not all, hematopoietic stem/progenitor and differentiated cell types (<https://www.haemosphere.org/expression/show?geneId=ENSMUSG00000000420>). Based on published knockout studies, this does not appear to be an essential gene for hematopoiesis. Panel c shows a series of tiny ChIP-seq traces, which are too small for rigorous evaluation, signal/noise does not appear to be high for multiple parameters, and positive and negative controls are not provided. Is this the best example of a potentially interesting gene that emerged from studying thousands of prospective cis-regulatory elements in the transgene context?

5) Fig. 8 – The model is very busy and mostly summarizes established old concepts from multiple investigators in the field: VEGF-Flk1 work from Choi, Rossant and others, AP1 work by multiple investigators, genes implicated in hemogenic endothelium function etc.

Reviewer #2 (Remarks to the Author):

Summary:

In this study, Edington-White and Maytum et al followed an impressive high-throughput technique to identify open chromatin regions that are also transcriptionally active by coupling ATAC-seq with a fluorescent-based reporter assay. Using this strategy, they identified functionally active DNA regulatory elements that control stage-specific transcription during gradual transition of mouse embryonic stem cells into hematopoietic progenitor cells (HSPCs) through hemogenic endothelium. Along with this technique, by integrating a serum-free in vitro differentiation system that relies on sequential addition of hematopoietic cytokines, e.g. BMP4, Activin, VEGF, SCF and interleukins, the authors attempted to identify the stimulation-responsive enhancers that are at play during this transition. They first tested how removal of individual growth factors affected the formation of hematopoietic progenitors (HPs) from hemogenic endothelium (HE) in comparison to the all-cytokine condition. Subsequently, comparing the ATAC-reporter profiles under the same conditions, they tried to pin-point the specific cytokine-responsive enhancers that regulate this transition. These results helped them identify VEGF-dependent enhancers that maintain the balance between endothelial to hematopoietic transition by controlling transcription. Upon identifying specific transcription factor motifs that are enriched in these VEGF-target enhancers, the authors claimed that VEGF response involves transcriptional regulation by TEAD and AP1 transcription factors that eventually inhibits upregulation of RUNX1 gene via upregulation of Notch1 and Sox17.

Identification of enhancer repertoires has been reported extensively during differentiation of human HSPCs into multiple blood lineages. However, this information is lacking for stages that are further upstream, e.g. during specification of hematopoietic stem cells from hemogenic endothelium. In that regard, this is an important study trying to identify the functionally important enhancers that are important for HSC specification. Classification of VEGF-responsive enhancer elements will also help understand the in-depth transcriptional mechanisms that this well-known cytokine plays to maintain balance between HE-HSC stages. Hence, this study will be really informative and attractive for HSC field and should be appreciated by a broad spectrum of audiences of transcriptional regulation of gene regulation. After answering the following questions, the manuscript could be a nice addition to "Nature Communications".

Comments:

1. For Fig. 1 and Fig. S1, the high-throughput reporter assay that the authors have utilized is unique, however, it is not clear how many ATAC-seq read(s) each cell is incorporating during the transfection event of reporter plasmid. Many regulatory elements work together

in three dimensional space by physically interacting with each other. Hence, if only one regulatory element is getting incorporated into cells per transfection, the authors may miss a lot of cis-elements that would work together in this apparently high-throughput method. This is important since this could likely impact their subsequent TF motif enrichment analyses that depend on the enhancers identified through this reporter approach. Can the authors provide an explanation for this?

2. In this assay, the reporter construct is integrated into a defined target site in the HPRT locus in the genome that is in general open and active in transcription. Can this influence the activity of ATAC-seq reads that are repressive in vivo but would be reflected as active in this assay because of the positional effect?

3. The correlation studies between the H3K27ac ChIP-seq data with the ATAC-seq based reporter assay showed a huge variation based on the stage of differentiation (30-60%, Fig 2a). Can the authors explain this difference? Also, did the authors overlap the stage-specific ATAC-seq peaks with the cis-regulatory elements that they found active in their reporter assay from each stage of differentiation? It would be good to know the percentage of overlap between these two datasets.

4. The authors should also correlate the RNA-seq based gene expression with the cis-elements identified with their ATAC-based reporter assay and also with the open chromatin regions identified through ATAC-seq for individual differentiation stages. It appears that the cis-elements identified through ATAC-based reporter assay should overlap better with the actual gene expression compared to the open chromatin elements identified by regular ATAC-seq. It would be nice to check if it is true. For this purpose, the authors could use either "nearest gene" approach or "Hi-C-based" approach to associate the genes with the cis-elements that potentially regulate their expression.

5. It appears that the authors have used two individual replicates for the ATAC-based reporter assays and went ahead with the cis-elements that appeared in both replicates (Fig. S1e). However, there is a huge difference in both the number of enhancer positive distal ATAC sites and the number of promoter ATAC sites in these two replicates (each differentiation stage in Rep 1 identified almost 10-fold more ATAC sites compared to Rep 2). This could bring in a lot of variation in the downstream analyses. Can the authors explain this mismatch and provide an explanation for the issues that may arise due to this?

6. Many times it appears that the manuscript lacks proper explanation for the figures and the analyses performed. Given the nature of the study the analyses are complicated and without proper explanation it is hard to appreciate the impact of these outcomes. For example,

Line 120: "To our surprise, most ATAC-fragments displaying stimulatory activity (see scheme in Figure S2a, Figure S1d, left panel) overlapped with ATAC-sites containing negatively scoring fragments, indicating that the vast majority of captured open chromatin regions can regulate transcription": It is not clear what is "ATAC-fragments displaying stimulatory activity" and what is "ATAC-sites containing negatively scoring fragments". From the terminology, it appears that both these fragments are determined from their ATAC-based reporter assay. Is it true? Also, Figure S1d, left panel should be bottom panel as the figure does not have any left panel.

Line 207: "The comparison of the chromatin signature of cells differentiated in serum and under serum-free conditions (Figure 4a), showed that around 80% of cis-elements seen in cells from serum-free culture overlapped in both conditions, demonstrating the reproducibility of our differentiation system. However, for HP cells we noticed changes in the bulk open chromatin landscape which affected the distal elements and not the promoters. This result indicates that although the cellular identity seemed to be largely preserved in sorted cells expressing the right combination of surface markers, the difference in the signalling environment exerted a strong effect on the chromatin landscape.": It is hard to determine the figure(s) from which the authors are making these conclusions. I would suggest the authors to point out the exact figures that they are referring while deducing a conclusion.

Line 221: "We next examined, which TF binding motifs were enriched in cytokine-responsive ATAC-Seq peaks harbouring enhancer activity (Figure 4b - d). Alteration of cytokine conditions had a profound influence on chromatin programming. The absence of BMP was incompatible with HP formation and open chromatin regions with enhancer activity containing SMAD, HOX, RAR and NOTCH motif signatures were lost in HE. This finding is in keeping with these factors being required to form the HE. We noticed that the presence of VEGF led to a loss of peaks with a hematopoietic motif signature in HE2/HP, such as RUNX1, FLI1, GATA and PU.1 motifs."

Line 247: "To identify VEGF responsive TFs, we conducted a supervised motif clustering analysis that highlighted cell type specific motif enrichments (Figure 5a-c). This analysis indicates that the withdrawal of VEGF activates enhancers with a hematopoietic motif signature with RUNX1 and PU.1 motifs. HP cells in VEGF cultures maintain an enrichment of motifs for RPBj and HES1 which are mediators of NOTCH signalling. Moreover, in contrast to +VEGF cultures, ATAC peaks in -VEGF HE1 cells were strongly enriched in TEAD motifs together with binding motifs for factors linked to inflammatory signalling (AP-1, NFkB and CREB/ATF) which has been shown to be important for stem cell development."

In both these examples, the authors essentially mention the whole figure once in the very first sentence and then come to a set of important conclusions in the following sentences without referring the specific panels of the figures that support these results. These are complicated analyses and without these explanations, it is very hard to follow the authors' claims.

Line 267: "Our ChIP data show that AP-1 and TEAD indeed bind to such motifs in the HE20 (Figure 6c) but once HP cells have formed, TEAD binding is lost.": In this case the authors didn't follow the proper order in which the specific panels of the figures should appear. Here, panel c for Fig. 6 appeared already without the mention of panels a and b before. This interrupts the flow of the manuscript.

Figure 5a: To show the effect of removal of individual cytokines on each stage of differentiation, the authors mention the specific growth factor that was removed as, "VEGF", "IL3", "IL6", "BMP4". To be clear, they should mention them with negative/delta signs as "-VEGF", "-IL3", "-IL6", "-BMP4". In fact, they have used this format for Figure 6d and they should be consistent throughout the manuscript.

7. The most elegant aspect of the strategy followed in this manuscript is to identify the stage-specific functional enhancers in a high-throughput setting. However, in multiple occasions, the authors have provided only one example to validate their genome-wide findings. For example, in line 117, the authors show the PU.1/SPI1 locus to support the validity of their ATAC-seq based reporter approach. The authors should provide additional known example loci that corroborate this analysis. To determine how response to cytokines regulates the activity of signaling-responsive enhancers during the transition of hemangioblasts to hematopoietic progenitors, the authors used the Galnt1 enhancer that binds the relevant transcription factors. There must be additional enhancers that follow these trends according to authors' hypothesis and they should use them in similar analyses to strengthen their model in a genome-wide level.

8. Over the past few years, the importance of stress/stimulation-responsive enhancers in human development and diseases have been implicated including the field of hematopoiesis. The authors should refer these studies in the relevant sections of the manuscript. For example,

Lines 196-203, this paragraph should include the following articles, one of which studied the inflammatory signaling responsive enhancer that regulate auto-immune disease and the other one identified an anemia-induced, stress-regulated enhancer that is active during stress hematopoiesis:

Simeonov, D. R. et al. Discovery of stimulation-responsive immune enhancers with CRISPR activation. Nature 549, 111-115, doi:10.1038/nature23875 (2017). PMC5675716

Hewitt, K. J. et al. GATA Factor-Regulated Samd14 Enhancer Confers Red Blood Cell Regeneration and Survival in Severe Anemia. Dev Cell 42, 213-225 e214,

doi:10.1016/j.devcel.2017.07.009 (2017). PMC5578808

Line 187: "Here, PU.1 motifs show increased co-localization with SMAD and OCT motifs (Figure 3e)." This observation is well-supported by the previous studies that showed signaling transcription factors, e.g. TGF β and BMP-induced SMADs, often follow lineage-specific master transcription factors to cell stage-specific enhancers. The authors should mention these studies:

Mullen, A. C. et al. Master transcription factors determine cell-type-specific responses to TGF-beta signaling. *Cell* 147, 565-576, doi:10.1016/j.cell.2011.08.050 (2011). PMC3212730

Trompouki, E. et al. Lineage regulators direct BMP and Wnt pathways to cell-specific programs during differentiation and regeneration. *Cell* 147, 577-589, doi:10.1016/j.cell.2011.09.044 (2011). PMC3219441

Line 222: "Alteration of cytokine conditions had a profound influence on chromatin programming. The absence of BMP was incompatible with HP formation and open chromatin regions with enhancer activity containing SMAD, HOX, RAR and NOTCH motif signatures were lost in HE." This finding is nicely supported by a previous study that showed the importance of BMP signaling in human CD34+ HSPCs and further suggested that the BMP-target enhancers often operate as transcriptional signaling centers since they are the docking sites for several other crucial signaling transcription factors. DNA-binding motifs of numerous signaling TFs within those BMP-targeted signaling centers, e.g. SMAD, GLI, HES, RAR, are disrupted by human red blood cell trait associated SNPs, suggesting crucial role of stimulation-responsive enhancers in hematopoietic traits and diseases. This study should be referenced as well:

Choudhuri, A. et al. Common variants in signaling transcription-factor-binding sites drive phenotypic variability in red blood cell traits. *Nat Genet* 52, 1333-1345, doi:10.1038/s41588-020-00738-2 (2020). PMC7876911

Reviewer #3 (Remarks to the Author):

In this manuscript, Edginton-White et al first developed a single-locus integrated reporter system to test the ability of ATAC-Seq fragments to drive reporter expression. They applied this approach to mouse embryonic stem cells differentiating towards hematopoietic progenitors and identified thousands of distal/proximal reporter positive sites. These sites are consistent with expected features of active enhancers. Using ATAC-Seq, the authors identify cytokine-dependent enhancers and their enriched transcription factor motifs. They show that loss of VEGF dramatically increases progenitor populations, and identify VEGF-responsive enhancers and motifs. The authors then dissect a Galnt1-proximal enhancer that integrates cytokine signals through associated transcription factors. Finally, the authors performed single-cell RNA-Seq to show that loss of VEGF results in increased Runx1 expression in progenitors, perhaps driven by increased chromatin accessibility at Runx1-proximal enhancers.

Overall, my impression is that this manuscript is trying to do too much, which confuses the main message of the work. The reporter assay section seems distinct from the cytokine genomics section. I would suggest restructuring the work to better integrate these two parts. In addition, while the text is mostly well-written, there are numerous instances in which the conclusions drawn are not clearly supported by the data presented (see below).

Major:

1. The complexity of plasmid libraries and integrated cellular libraries are important considerations of this technology, and should be documented. Please specify the total number of ATAC-Seq peaks in each cell type, the complexity of the cloned fragment library, and the median number of positive fragments per positive distal ATAC site.

2. It will be important to highlight the potential pitfalls of the endogenous reporter approach in the discussion section. In particular, please comment on the efficiency of the reporter integration into the HPRT locus and the complexity/coverage of the resulting cellular library. Also, why are there more promoter positive fragments than distal fragments identified in Supplemental Figure 1? The low efficiency of locus-specific integration, combined with the bias of ATAC-Seq fragments to promoters are drawbacks of the technology that should be discussed.

3. Can the authors estimate the saturation of enhancer/promoter positive regions? That is, what fraction of all enhancer/promoter positive regions were identified? Saturation analysis by downsampling positive fragments could help provide an estimate.

4. Figure legends are lacking sufficient detail to interpret experiments. For example, what do the tracks in Figure 1d represent? That is, are "enhancers" those defined by ATAC-Seq or reporter? Are they positive fragments? Are "Serum" tracks ATAC-Seq or reporter?

5. The authors write: "Importantly, our method recovers many enhancers described in the VISTA database, ..., but due to its sheer size, the dataset reported here vastly extends such enhancer sets." Since the VISTA database tests enhancers *in vivo*, this is not an appropriate comparison. More appropriate would be to compare with the number of previously performed reporter assays in cell culture, of which there are many.

6. The nomenclature of "+/-" cytokine is inconsistent and confusing. For example, Figure 4b is an overview of the motif analysis strategy in 4c/4d. It uses the phrase "- cytokine HE1", which I assume means loss of a given cytokine. However, Figure 4c labels like "+ VEGF" indicate that cytokines are being added, rather than removed. This makes it confusing to understand what experiment was actually performed. Also, in Figure 5a, please indicate that cytokines are being removed.

7. The authors write: "The absence of BMP was incompatible with HP formation and open chromatin regions with enhancer activity containing SMAD, HOX, RAR and NOTCH motif signatures were lost in HE." The result is not referenced as a figure. Figure 4c/d shows addition rather than absence of BMP.

8. The authors write: "We noticed that the presence of VEGF led to a loss of peaks with a hematopoietic motif signature in HE2/HP, such as RUNX1, FLI1, GATA and PU.1 motifs." This is not immediately clear from Figure 4c. What is the z score? Is this statistically significant? Similarly, the authors write: "HP cells in VEGF cultures maintain an enrichment of motifs for RPBj and HES1." The HES1 result is not apparent from the figure.

9. The authors write: "mutation of the TEAD binding sites led to an increase in reporter activity in HE1 cells, suggesting that here TEAD restricts enhancer activity." This is not clear from Figure 6d.

10. The authors refer to a "Galnt1 enhancer", but do not show that the enhancer regulates Galnt1 expression. Perhaps a more appropriate name is "Galnt1-proximal" enhancer. In addition, Figure 6d seems to show that different cytokine conditions yield similar cell type reporter results, indicating that this enhancer may not be the best example of a cytokine signal integrator.

Minor:

1. Typos throughout the manuscript, especially in figures. Figure 6: "H3K17ac". Figure 2:

"overlapping". Figure 4: "specific cytokines were withdrawn or left out at the beginning of blast culture."

2. Figure 5c-d are unintuitive compared to Figure S5c-e

Response to Reviewer comments

We thank the reviewers for their thoughtful and constructive suggestions which led to new findings and made the paper much better. Note that all changes in the manuscript are highlighted in red.

Reviewer #1 (Remarks to the Author):

The manuscript by Edginton-White describes a genomic effort to discover cis-regulatory elements operational during hematopoietic differentiation of mouse embryonic stem cells. The authors indicate that the analysis yielded “thousands of differentially active cis-elements” at multiple stages of differentiation. A major conclusion highlighted was that stage-specific cis-elements respond to cytokine signals that impact activity of signal-dependent transcription factors.

Response:

We apologize if we created the impression that we were claiming to be the first to report the presence of signalling-responsive cis-regulatory elements. Of course, we were not. We have gone through the text again to make sure that the cause for such a misunderstanding is eliminated and have toned down overenthusiastic statements in the abstract and the introduction. However, we were indeed surprised to learn from our experiments the scale of responsiveness with thousands of elements responding to the withdrawal of different cytokines. As a resource we provide lists of signalling responsive cis-regulatory elements for each cytokine assessed and provided insight into the transcription factor motifs enriched in each set of these elements, thus indicating which transcription factors are signalling responsive to each specific cytokine.

Reviewer #1

It was stated that the work “presents a major advance in our understanding of developmental gene expression”. The problem of how large numbers of cis-regulatory elements operate and are regulated during cellular transitions has obvious importance and is studied by many investigators. There have been many contributions to the problem, and the authors have selectively cited literature on this problem. There are important limitations of the current study.

Response:

We agree with the referee that a very large number of investigators have studied this question for a very long time (including the senior authors). We therefore opted to highlight the difficulties and problems by citing reviews rather than provide a comprehensive scholarly overview about the different strategies which is impossible in 4000 words. However, we have added a few more references to the text. In addition, we discuss the advantages and disadvantages of our method in Supplementary Notes as compared to others.

Reviewer #1

1) While the overall approach utilised has potential to yield significant insights, the major conclusions recapitulate known concepts. Multiple groups have described signal-dependent (cytokine, stress etc.) enhancers controlled by signal-dependent transcription factors (e.g. Choudhuri,,,, Zon Nature Genetics and multiple prior examples). This is a common paradigm that has been studied at the gene-specific level and via genomic approaches. Signal-dependent transcription factors often co-localize with hematopoietic transcription factors at enhancers, illustrating how cellular signalling mechanisms impact genome function. Overall, it is not at all evident what the “major advance” is.

Response:

Again, we apologize for giving the impression that we were the first people who discovered signalling responsive gene expression regulation. However, we still think that our work is a major advance for the field, both as a resource and a new methodology that can be expanded into other cell types. We could have written a Methods paper, but we felt that we needed to prove the usefulness of our method, hence the second part demonstrating that it is functionally characterized enhancer (and promoter) elements that respond to cytokine withdrawal. Most major journals request such proof and I believe that Nature Communications is no different in that respect.

Reviewer #1

2) A second major limitation relates to the argument presented in the introduction, highlighting the limitations of reporter gene assays to define cis-regulatory elements and the challenges of discovering elements operational in chromatin. A key point that is not addressed is that years of studies have demonstrated the importance of analysing cis-regulatory element function at endogenous loci and not in the context of transgenes integrated at ectopic sites. At the very least, discoveries utilising transgenes need to be validated at endogenous loci, and many prior studies have indicated that activity at ectopic loci does not predict endogenous locus activity – quantitatively and even qualitatively. Even with the early studies on the beta-globin locus, hypersensitive sites and cis-regulatory elements within them almost always were important at ectopic chromatin sites (e.g. linked to reporters, analogous to the author’s system), but careful analyses at endogenous loci demonstrated the promiscuity of elements at ectopic sites, which were often not essential at the endogenous locus. These studies have been extended well beyond the globin genes. Thus, to bolster arguments that this reporter-based strategy has unique utility for discovering important (or essential) cis-regulatory elements and that the study has yielded a valuable “resource”, data needs to be provided that tests cis-regulatory element function at the respective endogenous loci.

Response:

If we created the impression that our method quantitatively predicts enhancer activity at an endogenous locus, we again apologise. What our method does, is to identify elements that are capable of stimulating a minimal promoter, just as thousands of transient transfection reporter gene assays have done in the past and which have operationally defined enhancer elements. We can indeed not measure quantitative contribution of a single element within a single locus. This can indeed only be done by genomic editing but to perform this technique in a high throughput fashion for a differentiating system is challenging.

However, our method presents several very important advances: (i) We test global enhancer function in a chromatin context, (ii) we measure global enhancer activity in a safe harbour site thus largely eliminating position effects and, most importantly, (iii) we can follow the activity of a single element throughout a differentiation pathway in the same experiment and link this activity with the presence and absence of specific transcription factors and factor binding sites. I do not recall a single genome-wide method that presents such features. Our new data about the developmental regulation of different enhancers analysed in isolation show what can be done.

Our new resource informs (i) where to find candidate elements for further studies, (ii) what transcription factors they interact with, and (iii) whether a respective element is cytokine responsive, the way it responds, and which transcription factors may associate with responsiveness. Moreover, the integration of our data with previous multi-omics analyses (including ChIP, gene expression and Hi-C studies) and the ability to rapidly (within 3-4 weeks) test mutant elements allows to characterise the newly discovered cis-regulatory elements in exquisite detail throughout an entire differentiation pathway. We stand by our statement that this is a major advance for the field.

To satisfy the reviewer, we have gone through the literature and have identified cis-regulatory elements identified by our resource which have been studied by gene targeting experiments of endogenous loci. These include the Tal1 -40kb enhancer, the +9.5 Gata2 enhancer, the -14kb PU.1 enhancer, numerous elements of the Sox2 “super enhancer” and multiple enhancers from the ES cell stage identified by a CRISPR screen. We therefore did not feel that we had to further validate our method.

However, note that (i) we can connect enhancer activity to the activity of their associate genes; (ii) we were able to recover individual elements and validate their predicted activity which we have now included in Figure 2 and S2, (iii) use such elements to identify the transcription factors contributing to their activity, (iv) identify which elements are developmentally regulated and (v) which ones respond to signalling (New Figs 7, 8 and Supplementary Notes). Our resource provides ample data to further validate these elements by mutagenesis and genomic editing. As requested, we have added a number of examples where we manually validated single elements and correlated their activity to the presence or absence of a DHS (Fig 2 f - h; Fig S2 g).

Reviewer #1

Additional Comments:

1) Methodological details – Fig. 1 legend and text in Results do not clearly describe the high-throughput strategy. There was little to no discussion of issues related to quantitation, statistical analyses, limitations of using the reporter system etc.

Response:

We agree with the reviewer that there should be more discussion of the methodological details. We have therefore expanded the main text to provide further explanations, added to the Methods section and created a Supplementary Discussion section in the Supplementary materials that answers the questions.

2) Fig. 2 – “ocerlapping”

Response: This typo was corrected

3) Legends commonly lacked statistical strategies, and most figures lacked this essential analytical information.

Response: We apologize for the omission. This deficiency has been corrected. Statistical analyses have been added in the figures, the main text, and the legends wherever appropriate. In addition, as advised by a Senior Bioinformatician we increased the statistical robustness of our motif enrichment analyses in Figures 4, S4, 5 and S5 by performing only motif enrichment analysis at peaks that differed more than two-fold between the cytokine and non-cytokine condition instead of using simple peak overlaps. This required to extensively re-analyse a number of integrated data-sets, but it was worth it. The outcome of this analysis is the same as in previous analyses (VEGF blocks the activation of elements with motifs for hematopoietic transcription factors) but stands on much better statistical ground. In addition, we further curated the motifs and eliminated redundant motifs. This strategy cleared up minor inconsistencies between the motif analyses shown in Figure 5 and S5.

4) Fig. 6 – The authors were interested in *Galnt1*. *Galnt1* is a broadly expressed gene, with mRNA present in most, if not all, hematopoietic stem/progenitor and differentiated cell types (<https://www.haemosphere.org/expression/show?geneId=ENSMUSG00000000420>). Based on published knockout studies, this does not appear to be an essential gene for hematopoiesis. Panel c shows a series of tiny ChIP-seq traces, which are too small for rigorous evaluation, signal/noise does not appear to be high for multiple parameters, and positive and negative controls are not provided. Is this the best example of a potentially interesting gene that emerged from studying thousands of prospective cis-regulatory elements in the transgene context?

Response:

We acknowledge that this gene is not the most obvious reporter to test cytokine responsiveness. We chose it not because we were interested in the gene, but because we wanted to link cytokine responsiveness to specific TFs using a mutagenesis strategy. The criteria for choosing it were: i) it is VEGF responsive, (ii) it has binding sites for the TFs we noted to be involved in regulating cis-element activity such as TEAD, RUNX1, AP-1 etc. (iii) expression is dependent on the AP-1 TF family as determined by previous studies.

To satisfy the reviewer, we have now included additional VEGF-responsive elements from our enhancer collection. Importantly, we examined the main +23 kb enhancer of the *Runx1* locus. We also show the responsiveness of other elements such as the *Dlk1* enhancer, Fig 8c or the *Hspg2*, *Pxn* and *Sparc* enhancers. We also provide the sequences with annotated motifs in the Supplementary Notes to show that such elements show a TEAD / RUNX1 / SOX signature. Most importantly, we have performed mutagenesis analyses of the =23 kb *RUNX1* enhancer to confirm our hypothesis that the axis TEAD and RUNX1 regulates the activity of *RUNX1* during the EHT. Note that the latter result adds a so far unknown aspect to the regulation of the EHT by RUNX1.

We therefore removed some of the analyses of the *Galnt1* enhancer, in particular the examination of its activity in the presence and absence of other cytokines and just retained those removing VEGF. The removed data will be available in Alex Maytum's thesis. We agree that the *Galnt1* browser screenshot in Figure 6 was of substandard quality and have removed

it. To enable the reviewers to zoom in and see the data in more detail we have generated a Browser link for the reviewers showing the raw reads that shows all chromatin and ChIP data linked to the identified enhancer elements.

<https://genome-euro.ucsc.edu/s/b.edginton%2Dwhite/EdgintonWhiteetal2022>

We also added screenshots showing ChIP experiments where we zoomed in for the different enhancer elements. We kept the screenshots in the Supplementary Notes to provide an overview. For the final publication, and in addition to depositing the data at GEO we have generated a more stable Track-Hub (for details see resource summary).

5) Fig. 8 – The model is very busy and mostly summarizes established old concepts from multiple investigators in the field: VEGF-Flk1 work from Choi, Rossant and others, AP1 work by multiple investigators, genes implicated in hemogenic endothelium function etc.

Response: This notion is of course correct, and we had already acknowledged this fact by saying “Although most of the network components and their different roles are known from many perturbation and knock-out experiments, we have now identified the TF involved and the elements upon which they act, providing a rich resource for studies of identifying the signals required for the activation of the correct gene expression program required for efficient blood cell production.

We hope that we have made this point clearer. We also have made the model easier to follow by splitting it into three panels.

Reviewer #2 (Remarks to the Author):

In this study, Edington-White and Maytum et al followed an impressive high-throughput technique to identify open chromatin regions that are also transcriptionally active by coupling ATAC-seq with a fluorescent-based reporter assay. Using this strategy, they identified functionally active DNA regulatory elements that control stage-specific transcription during gradual transition of mouse embryonic stem cells into hematopoietic progenitor cells (HSPCs) through hemogenic endothelium. Along with this technique, by integrating a serum-free in vitro differentiation system that relies on sequential addition of hematopoietic cytokines, e.g. BMP4, Activin, VEGF, SCF and interleukins, the authors attempted to identify the stimulation-responsive enhancers that are at play during this transition. They first tested how removal of individual growth factors affected the formation of hematopoietic progenitors (HPs) from hemogenic endothelium (HE) in comparison to the all-cytokine condition.

Subsequently, comparing the ATAC-reporter profiles under the same conditions, they tried to pin-point the specific cytokine-responsive enhancers that regulate this transition. These results helped them identify VEGF-dependent enhancers that maintain the balance between endothelial to hematopoietic transition by controlling transcription. Upon identifying specific transcription factor motifs that are enriched in these VEGF-target enhancers, the authors claimed that VEGF response involves transcriptional regulation by TEAD and AP1 transcription

factors that eventually inhibits upregulation of RUNX1 gene via upregulation of Notch1 and Sox17.

Identification of enhancer repertoires has been reported extensively during differentiation of human HSPCs into multiple blood lineages. However, this information is lacking for stages that are further upstream, e.g. during specification of hematopoietic stem cells from hemogenic endothelium. In that regard, this is an important study trying to identify the functionally important enhancers that are important for HSC specification. Classification of VEGF-responsive enhancer elements will also help understand the in-depth transcriptional mechanisms that this well-known cytokine plays to maintain balance between HE-HSC stages. This study will be really informative and attractive for HSC field and should be appreciated by a broad spectrum of audiences of transcriptional regulation of gene regulation. After answering the following questions, the manuscript could be a nice addition to “Nature Communications”.

Response: We thank the reviewer for their enthusiasm

Comments:

1. For Fig. 1 and Fig. S1, the high-throughput reporter assay that the authors have utilized is unique, however, it is not clear how many ATAC-seq read(s) each cell is incorporating during the transfection event of reporter plasmid.

Response: We have now added a sentence saying that just one element is integrated per cell. The HPRT locus is located on the X-chromosome and only one allele is active.

Many regulatory elements work together in three-dimensional space by physically interacting with each other. Hence, if only one regulatory element is getting incorporated into cells per transfection, the authors may miss a lot of cis-elements that would work together in this apparently high-throughput method. This is important since this could likely impact their subsequent TF motif enrichment analyses that depend on the enhancers identified through this reporter approach. Can the authors provide an explanation for this?

Response: Our method measures whether an element can stimulate a promoter just as in many other reporter assays before which defined enhancers. We therefore can only measure the contribution of the motifs in these elements and correlate their presence or absence with the developmental activity of this single element and whether the DHS marking its activity within the whole locus is there or not. We are of course aware that a full locus is regulated in a complex fashion. See our response to Reviewer 1 to this regard. However, note that in all cases we examined, the presence or absence of a DHS at the endogenous locus is predictive for enhancer activity.

2. In this assay, the reporter construct is integrated into a defined target site in the HPRT locus in the genome that is in general open and active in transcription. Can this influence the activity of ATAC-seq reads that are repressive in vivo but would be reflected as active in this assay because of the positional effect?

Response: We wanted to identify enhancer. i.e. stimulatory elements. Repressive elements, i.e. elements that recruit co-repressors, will not score in our assay as the minimal promoter

has no or little activity. However, note that for individual loci the method allows to integrate larger fragments and thus study cis-element cooperation in development.

3. The correlation studies between the H3K27ac ChIP-seq data with the ATAC-seq based reporter assay showed a huge variation based on the stage of differentiation (30-60%, Fig 2a). Can the authors explain this difference?

Response: We assume that this is due to the quality of the ChIP data. Note that the ES cell data are from public sources, our own data (HB - HP) show much less variation.

Also, did the authors overlap the stage-specific ATAC-seq peaks with the cis-regulatory elements that they found active in their reporter assay from each stage of differentiation? It would be good to know the percentage of overlap between these two datasets.

Response: We thank the reviewer for this suggestion. We have included such data in the revision. We find that between 30% and 50% of all distal ATAC sites and nearly 80% of all promoter ATAC sites show stimulatory activity (New Figure 2a, New Figure S2b), see also the saturation curve below.

4. The authors should also correlate the RNA-seq based gene expression with the cis-elements identified with their ATAC-based reporter assay and also with the open chromatin regions identified through ATAC-seq for individual differentiation stages. It appears that the cis-elements identified through ATAC-based reporter assay should overlap better with the actual gene expression compared to the open chromatin elements identified by regular ATAC-seq. It would be nice to check if it is true. For this purpose, the authors could use either “nearest gene” approach or “Hi-C-based” approach to associate the genes with the cis-elements that potentially regulate their expression.

Response: We apologise if this was not clear. This is precisely what we have done, and the result is shown in Figure 2e. Enhancer activity and gene expression are well correlated. We have added a sentence to explain better what we have done.

5. It appears that the authors have used two individual replicates for the ATAC-based reporter assays and went ahead with the cis-elements that appeared in both replicates (Fig. S1e). However, there is a huge difference in both the number of enhancer positive distal ATAC sites and the number of promoter ATAC sites in these two replicates (each differentiation stage in Rep 1 identified almost 10-fold more ATAC sites compared to Rep 2). This could bring in a lot of variation in the downstream analyses. Can the authors explain this mismatch and provide an explanation for the issues that may arise due to this?

Response: We agree that the number of elements discovered in the two assays is different, but the smaller set is contained within the larger set. We disagree that this variability in numbers will increase variability and randomness in identifying elements, as all data have been extensively filtered against numerous criteria as described in methods. The only thing that may happen is that we miss elements. However, our coverage with regards to ATAC sequences is excellent as shown in Figures 2a and S2b (see also the saturation analysis below).

To further elaborate on this issue, we have included a "Troubleshooting" chapter in the Supplementary Notes with an explanation.

6. Many times it appears that the manuscript lacks proper explanation for the figures and the analyses performed. Given the nature of the study the analyses are complicated and without proper explanation it is hard to appreciate the impact of these outcomes. For example,

Line 120: "To our surprise, most ATAC-fragments displaying stimulatory activity (see scheme in Figure S2a, Figure S1d, left panel) overlapped with ATAC-sites containing negatively scoring fragments, indicating that the vast majority of captured open chromatin regions can regulate transcription": It is not clear what is "ATAC-fragments displaying stimulatory activity" and what is "ATAC-sites containing negatively scoring fragments". From the terminology, it appears that both these fragments are determined from their ATAC-based reporter assay. Is it true? Also, Figure S1d, left panel should be bottom panel as the figure does not have any left panel.

Response: We apologise for the confusion. We have rephrased the sentences to make this clearer

Line 207: "The comparison of the chromatin signature of cells differentiated in serum and under serum-free conditions (Figure 4a), showed that around 80% of cis-elements seen in cells from serum-free culture overlapped in both conditions, demonstrating the reproducibility of our differentiation system. However, for HP cells we noticed changes in the bulk open chromatin landscape which affected the distal elements and not the promoters. This result indicates that although the cellular identity seemed to be largely preserved in sorted cells expressing the right combination of surface markers, the difference in the signalling environment exerted a strong effect on the chromatin landscape.": It is hard to determine the figure(s) from which the authors are making these conclusions. I would suggest the authors to point out the exact figures that they are referring while deducing a conclusion.

Response: We apologise for the confusion – this figure (Fif.S4a) was indeed not mentioned. We have added this information.

Line 221: "We next examined, which TF binding motifs were enriched in cytokine-responsive ATAC-Seq peaks harbouring enhancer activity (Figure 4b - d). Alteration of cytokine conditions had a profound influence on chromatin programming. The absence of BMP was incompatible with HP formation and open chromatin regions with enhancer activity containing SMAD, HOX, RAR and NOTCH motif signatures were lost in HE. This finding is in keeping with these factors being required to form the HE. We noticed that the presence of VEGF led to a loss of peaks with a hematopoietic motif signature in HE2/HP, such as RUNX1, FLI1, GATA and PU.1 motifs."

Line 247: "To identify VEGF responsive TFs, we conducted a supervised motif clustering analysis that highlighted cell type specific motif enrichments (Figure 5a–c). This analysis indicates that the withdrawal of VEGF activates enhancers with a hematopoietic motif signature with RUNX1 and PU.1 motifs. HP cells in VEGF cultures maintain an enrichment of motifs for RPBj and HES1 which are mediators of NOTCH signalling. Moreover, in contrast to

+VEGF cultures, ATAC peaks in -VEGF HE1 cells were strongly enriched in TEAD motifs together with binding motifs for factors linked to inflammatory signalling (AP-1, NFkB and CREB/ATF) which has been shown to be important for stem cell development.”:

In both these examples, the authors essentially mention the whole figure once in the very first sentence and then come to a set of important conclusions in the following sentences without referring the specific panels of the figures that support these results. These are complicated analyses and without these explanations, it is very hard to follow the authors' claims.

Response: We apologise for the omission. We have added this information and explained what we have done.

Line 267: “Our ChIP data show that AP-1 and TEAD indeed bind to such motifs in the HE20 (Figure 6c) but once HP cells have formed, TEAD binding is lost.”: In this case the authors didn't follow the proper order in which the specific panels of the figures should appear. Here, panel c for Fig. 6 appeared already without the mention of panels a and b before. This interrupts the flow of the manuscript.

Response: We have completely rewritten this particular chapter in response to Reviewer 3 and hope that everything is clearer now.

Figure 5a: To show the effect of removal of individual cytokines on each stage of differentiation, the authors mention the specific growth factor that was removed as, “VEGF”, “IL3”, “IL6”, “BMP4”. To be clear, they should mention them with negative/delta signs as “-VEGF”, “-IL3”, “-IL6”, “-BMP4”. In fact, they have used this format for Figure 6d and they should be consistent throughout the manuscript.

Response: We apologise for the confusion. We have now unified our nomenclature.

7. The most elegant aspect of the strategy followed in this manuscript is to identify the stage-specific functional enhancers in a high-throughput setting. However, in multiple occasions, the authors have provided only one example to validate their genome-wide findings. For example, in line 117, the authors show the PU.1/SPI1 locus to support the validity of their ATAC-seq based reporter approach. The authors should provide additional known example loci that corroborate this analysis.

Response: We have gone one step further and provide browser and Track Hub links (see above and in the text) that allows everyone to look at their favourite gene and we have included more examples of enhancer analyses in isolation by creating a number of new ES cell lines.. We also added references that demonstrate that elements captured by our assay also show enhancer activity when deleted within the context of an endogenous locus.

To determine how response to cytokines regulates the activity of signaling-responsive enhancers during the transition of hemangioblasts to hematopoietic progenitors, the authors

used the Galnt1 enhancer that binds the relevant transcription factors. There must be additional enhancers that follow these trends according to authors' hypothesis and they should use them in similar analyses to strengthen their model in a genome-wide level.

Response: We did include additional examples. See our response to Referee 1

8. Over the past few years, the importance of stress/stimulation-responsive enhancers in human development and diseases have been implicated including the field of hematopoiesis. The authors should refer these studies in the relevant sections of the manuscript. For example,

Lines 196-203, this paragraph should include the following articles, one of which studied the inflammatory signaling responsive enhancer that regulate auto-immune disease and the other one identified an anemia-induced, stress-regulated enhancer that is active during stress hematopoiesis:

Simeonov, D. R. et al. Discovery of stimulation-responsive immune enhancers with CRISPR activation. *Nature* 549, 111-115, doi:10.1038/nature23875 (2017). PMC5675716

Hewitt, K. J. et al. GATA Factor-Regulated Samd14 Enhancer Confers Red Blood Cell Regeneration and Survival in Severe Anemia. *Dev Cell* 42, 213-225 e214, doi:10.1016/j.devcel.2017.07.009 (2017). PMC5578808

Line 187: "Here, PU.1 motifs show increased co-localization with SMAD and OCT motifs (Figure 3e)." This observation is well-supported by the previous studies that showed signaling transcription factors, e.g. TGF β and BMP-induced SMADs, often follow lineage-specific master transcription factors to cell stage-specific enhancers. The authors should mention these studies:

Mullen, A. C. et al. Master transcription factors determine cell-type-specific responses to TGF-beta signaling. *Cell* 147, 565-576, doi:10.1016/j.cell.2011.08.050 (2011). PMC3212730

Trompouki, E. et al. Lineage regulators direct BMP and Wnt pathways to cell-specific programs during differentiation and regeneration. *Cell* 147, 577-589, doi:10.1016/j.cell.2011.09.044 (2011). PMC3219441

Line 222: "Alteration of cytokine conditions had a profound influence on chromatin programming. The absence of BMP was incompatible with HP formation and open chromatin regions with enhancer activity containing SMAD, HOX, RAR and NOTCH motif signatures were lost in HE." This finding is nicely supported by a previous study that showed the importance of BMP signaling in human CD34+ HSPCs and further suggested that the BMP-target enhancers often operate as transcriptional signaling centers since they are the docking sites for several other crucial signaling transcription factors. DNA-binding motifs of numerous signaling TFs within those BMP-targeted signaling centers, e.g. SMAD, GLI, HES, RAR, are disrupted by human red blood cell trait associated SNPs, suggesting crucial role of stimulation-responsive enhancers in hematopoietic traits and diseases. This study should be referenced as well:

Choudhuri, A. et al. Common variants in signaling transcription-factor-binding sites drive phenotypic variability in red blood cell traits. *Nat Genet* 52, 1333-1345, doi:10.1038/s41588-020-00738-2 (2020). PMC7876911

Response: We thank the reviewer for these suggestions and have included these references in the text, with the exception of the last which, whilst a nice example of signalling-dependent TF cooperation did not really fit into the text. However, we realized that we had missed to add a reference to the sentence “This finding is in keeping with these factors being required to form the HE” where we now cite the appropriate literature.

Reviewer #3 (Remarks to the Author):

In this manuscript, Edginton-White et al first developed a single-locus integrated reporter system to test the ability of ATAC-Seq fragments to drive reporter expression. They applied this approach to mouse embryonic stem cells differentiating towards hematopoietic progenitors and identified thousands of distal/proximal reporter positive sites. These sites are consistent with expected features of active enhancers. Using ATAC-Seq, the authors identify cytokine-dependent enhancers and their enriched transcription factor motifs. They show that loss of VEGF dramatically increases progenitor populations, and identify VEGF-responsive enhancers and motifs. The authors then dissect a Galnt1-proximal enhancer that integrates cytokine signals through associated transcription factors. Finally, the authors performed single-cell RNA-Seq to show that loss of VEGF results in increased Runx1 expression in progenitors, perhaps driven by increased chromatin accessibility at Runx1-proximal enhancers.

Overall, my impression is that this manuscript is trying to do too much, which confuses the main message of the work. The reporter assay section seems distinct from the cytokine genomics section. I would suggest restructuring the work to better integrate these two parts. In addition, while the text is mostly well-written, there are numerous instances in which the conclusions drawn are not clearly supported by the data presented (see below).

Response: We have taken this comment to heart and have significantly restructured the paper, removed superfluous data and added new ones as outlined in the response to the other two reviewers.

Major:

1. The complexity of plasmid libraries and integrated cellular libraries are important considerations of this technology and should be documented. Please specify the total number of ATAC-Seq peaks in each cell type, the complexity of the cloned fragment library, and the median number of positive fragments per positive distal ATAC site.

Response: We have now provided these data in the manuscript in Figures 1, 2a, S1e and S2b.

2. It will be important to highlight the potential pitfalls of the endogenous reporter approach in the discussion section. In particular, please comment on the efficiency of the reporter integration into the HPRT locus and the complexity/coverage of the resulting cellular library. Also, why are there more promoter positive fragments than distal fragments identified in Supplemental Figure 1? The low efficiency of locus-specific integration, combined with the bias of ATAC-Seq fragments to promoters are drawbacks of the technology that should be discussed.

Response: We thank the reviewer for their suggestions. We have now included an analysis that shows the coverage of ATAC sites in both libraries (New Fig S1d). With regards to the overrepresentation of promoter sequences: The reason for this is most likely the fact that (i) a large number of promoters are of a ubiquitous nature and (ii) are usually bigger than distal elements - we regard all sequences as promoters that are within 1.5 kb of the transcription start site.

Drawbacks: As outlined above we have created an additional supplementary Discussion with troubleshooting tips. However, we don't see the overrepresentation of promoters as drawbacks. On the contrary, it showed us that our global coverage is excellent. It is very easy to filter out such sequences or refine the interpretation when looking at individual gene loci.

3. Can the authors estimate the saturation of enhancer/promoter positive regions? That is, what fraction of all enhancer/promoter positive regions were identified? Saturation analysis by down-sampling positive fragments could help provide an estimate.

Response: Again, a very good suggestion. The percentage of ATAC-sites containing stimulating fragments is shown in Figure 2a and in Figure S2b, for distal and promoter sites, respectively. We have performed a saturation analysis as requested and we show the data for the review only as the result does not differ to what is in the figures already. The coverage for promoter sites is excellent, we catch almost 90% of all sites and all have stimulatory activity, highlighting the general quality and coverage of our library. The data for distal elements show a lower and more variable coverage. This does not come as a surprise because not all ATAC-sites (such as CTCF sites) are enhancers or are primed sites that are open but do not yet stimulate transcription. We are very happy with this result as it makes biological sense. We are currently putting together a second paper where we examine the "priming" sites in more detail.

Saturation analysis performed by downsampling the number of positive enhancer screen fragments in promoter (a) and distal (b) ATAC sites at each differentiation stage. Shown as percentage of ATAC sites overlapping with at least one positive fragment.

4. Figure legends are lacking sufficient detail to interpret experiments. For example, what do the tracks in Figure 1d represent? That is, are "enhancers" those defined by ATAC-Seq or reporter? Are they positive fragments? Are "Serum" tracks ATAC-Seq or reporter?

Response: We apologize for this deficiency and sloppy language. I have conducted a solid overhaul of all legends to make them clearer

5. The authors write: "Importantly, our method recovers many enhancers described in the VISTA database, ..., but due to its sheer size, the dataset reported here vastly extends such enhancer sets." Since the VISTA database tests enhancers *in vivo*, this is not an appropriate comparison. More appropriate would be to compare with the number of previously performed reporter assays in cell culture, of which there are many.

Response: We respectfully disagree here. If an element carrying the same DNA sequence scores positive under much more stringent conditions *in vivo* and it does the same in our assay, we see this as a confirmation that its activity is real. However, to satisfy the referee, we removed the second part of this sentence.

6. The nomenclature of "+/-" cytokine is inconsistent and confusing. For example, Figure 4b is an overview of the motif analysis strategy in 4c/4d. It uses the phrase "- cytokine HE1", which I assume means loss of a given cytokine. However, Figure 4c labels like "+ VEGF" indicate that cytokines are being added, rather than removed. This makes it confusing to understand what experiment was actually performed. Also, in Figure 5a, please indicate that cytokines are being removed.

Response: We apologise for the confusion. We have now unified our nomenclature and added more explanations in the legends.

7. The authors write: "The absence of BMP was incompatible with HP formation and open chromatin regions with enhancer activity containing SMAD, HOX, RAR and NOTCH motif signatures were lost in HE." The result is not referenced as a figure. Figure 4c/d shows addition rather than absence of BMP.

Response: We apologise for the omission. This has now been rectified

8. The authors write: "We noticed that the presence of VEGF led to a loss of peaks with a hematopoietic motif signature in HE2/HP, such as RUNX1, FLI1, GATA and PU.1 motifs." This is not immediately clear from Figure 4c. What is the z score? Is this statistically significant?

Response: This is one of the figures where we omitted to say in which panel to look and we seriously apologize for the omission. This result is obvious from Fig.4d where we highlight peaks with enhancer activity that are *lower* with VEGF. The motifs in question are at the lower third of the heat map and are clearly marked red in HE2/HP.

Similarly, the authors write: "HP cells in VEGF cultures maintain an enrichment of motifs for RPBj and HES1." The HES1 result is not apparent from the figure.

Response: This is seen in Fig.4c, where we highlight peaks that are *higher* with VEGF. The RUNX1 motifs show up in blue (low) whereas the HES1/RBPj motifs show up in red in the HP.

9. The authors write: "mutation of the TEAD binding sites led to an increase in reporter activity in HE1 cells, suggesting that here TEAD restricts enhancer activity." This is not clear from Figure 6d.

Response: We are intensely grateful for this comment. As a result, we had a closer look at the enhancer sequence and discovered an overlapping RUNX motif on the opposite strand to the left of the TEAD motif which is also destroyed by mutagenesis. We were always struggling with this result, but this discovery and inspecting our ChIP data made everything clear. TEAD-AP-1 binds in the HE, RUNX1 goes up in HP cells and kicks it off. If the site is mutated, the latter does not happen and enhancer activity in the population (a continuum) goes up. Much more logical mechanism. As it turns out, we find a similar motif architecture at other RUNX1 repressed endothelial genes as well (see additional Figures).

10. The authors refer to a "Galnt1 enhancer", but do not show that the enhancer regulates Galnt1 expression. Perhaps a more appropriate name is "Galnt1-proximal" enhancer. In addition, Figure 6d seems to show that different cytokine conditions yield similar cell type reporter results, indicating that this enhancer may not be the best example of a cytokine signal integrator.

Response: We used publicly available Hi-C data to assign the enhancer (Wilson et al., 2016). In addition, the hypersensitive site marking this element closely follows gene expression. The closest gene to the element, *Ino80*, does not do that.

As outlined above in our response to Referee 1, we now only show the VEGF results for *Galnt1* and we have added several examples of other VEGF-responsive enhancers (such as the *Dlk1* enhancer, Fig 8c or the enhancers shown in the new Fig 6 a).

Minor:

1. Typos throughout the manuscript, especially in figures. Figure 6: "H3K17ac". Figure 2: "overlapping". Figure 4: "specific cytokines were withdrawn of left out at the beginning of blast culture."

Response: This error has been corrected.

2. Figure 5c-d are unintuitive compared to Figure S5c-e

Response: We provided Figure S5c-e to show that the motif enrichment as seen in the heatmaps are based on real differences in the ATAC data and that these data are of good quality. There was no way how we could have included all ATAC data from all cytokine conditions in the paper and therefore decided to use heatmaps.

We hope that our manuscript now fulfils the criteria to be published in Nature Communications.

Yours sincerely on behalf of all authors

Constanze Bonifer

REVIEWERS' COMMENTS

Reviewer #1 (Remarks to the Author):

The revisions have appropriately addressed many of the prior points. Although the limitations of high-throughput approaches that do not investigate mechanisms operational at endogenous loci were recognized by the authors, the work presented is well controlled and should be of interest to investigators in the field and perhaps more broadly.

Reviewer #2 (Remarks to the Author):

I like the revised version of this manuscript agree with authors' edits. I commend the authors for their hard work and congratulate them for a very nice piece of science.

Reviewer #3 (Remarks to the Author):

The authors have sufficiently addressed my critiques from the previous submission.

I have one extra comment. In Figure 3b-f, the heatmap regarding pairwise analysis is mostly symmetric, which makes sense. However, shouldn't the diagonals have a uniform score? I see diverse scores on the diagonal, which is confusing if I understand the heatmap correctly.

Reviewer 3 asked:

I have one extra comment. In Figure 3b-f, the heatmap regarding pairwise analysis is mostly symmetric, which makes sense. However, shouldn't the diagonals have a uniform score? I see diverse scores on the diagonal, which is confusing if I understand the heatmap correctly.

Answer: We do not expect the diagonal to be a uniform line. In the heatmap the diagonal represents the tendency for a motif to co-localize with itself which only yields a high score with groups of multiple of the same motif within 50bp of each other, and if these groups occur more frequently compared to the background peak set. This feature will be different for each motif.

We have added this explanation to the figure legend of Figure 3

We hope that this answers the questions

Best wishes

Constanze Bonifer